# Mechanisms of activation and desensitization of full-length glycine receptor in lipid nanodiscs

Arvind Kumar [1], Sandip Basak [1], Shanlin Rao[2], Yvonne Gicheru[1], Megan L. Mayer[3], Mark S. P. Sansom [2] & Sudha Chakrapani[1,4✉]

Glycinergic synapses play a central role in motor control and pain processing in the central nervous system. Glycine receptors (GlyRs) are key players in mediating fast inhibitory neurotransmission at these synapses. While previous high-resolution structures have provided insights into the molecular architecture of GlyR, several mechanistic questions pertaining to channel function are still unanswered. Here, we present Cryo-EM structures of the full-length GlyR protein complex reconstituted into lipid nanodiscs that are captured in the unliganded (closed), glycine-bound (open and desensitized), and allosteric modulator-bound conformations. A comparison of these states reveals global conformational changes underlying GlyR channel gating and modulation. The functional state assignments were validated by molecular dynamics simulations, and the observed permeation events are in agreement with the anion selectivity and conductance of GlyR. These studies provide the structural basis for gating, ion selectivity, and single-channel conductance properties of GlyR in a lipid environment.

[1] Department of Physiology and Biophysics, Case Western Reserve University, Cleveland, OH 44106-4970, USA. [2] Department of Biochemistry, University of Oxford, Oxford OX1 3QU, UK. [3] Division of CryoEM and Bioimaging, SSRL, SLAC National Accelerator Laboratory, Stanford University, Menlo Park, CA, 94025, USA. [4] Department of Neuroscience, School of Medicine, Case Western Reserve University, Cleveland, OH 44106-4970, USA. ✉email: Sudha.chakrapani@case.edu

Glycine receptors (GlyRs), along with γ-aminobutyric acid receptors (GABA$_A$Rs), are the principal determinants of fast inhibitory synaptic neurotransmission in the central nervous system (CNS). GlyR and GABA$_A$R belong to the superfamily of pentameric ligand-gated ion channels (pLGICs). The vertebrate members of pLGIC also include excitatory cation-selective channels (nicotinic acetylcholine receptors—nAChRs and serotonin receptors—5HT$_3$Rs). Glycinergic neurotransmission critically regulates motor coordination, sensory reflex activity, and respiratory rhythms. GlyR dysfunctions are associated with neuromotor deficiencies such as hyperekplexia and epilepsy, and underlie chronic inflammatory pain[1,2]. Agents that potentiate GlyR function produce analgesia making them attractive candidates for pain therapy[3]. GlyRs assemble as either homopentamers of α subunits (α1−α4) or heteropentamers of α and β subunits. GlyR has a conserved pLGIC architecture consisting of an extracellular domain (ECD), a transmembrane domain (TMD), and an intracellular domain (ICD). The ECD houses the neurotransmitter binding pocket and the TMD governs the machinery for selective ion permeation. The ICD is important for receptor trafficking, synaptic clustering, and a central site for posttranslational modifications and regulation by endogenous and exogenous modulators[4]. Broadly, pLGIC function involves transitions between at least three types of distinct functional states, namely the resting, open, and the desensitized states. Upon agonist binding, the channel switches from the resting (Apo) state to open and desensitized conformations. Allosteric ligands modulate pLGIC activity by shifting the equilibrium between these functional states.

Recently, there has been ground-breaking progress in structure determination of both cationic and anionic members of the pLGIC family, providing a detailed view of assembly and conformational changes in homomeric and heteromeric channels[5–11]. For GlyRs, X-ray crystallographic and cryo-electron microscopic (cryo-EM) studies have provided the first snapshots of the channel in distinct conformations[8,11]. These landmark studies revealed a conserved mechanism for channel activation involving an agonist-induced global twist in the ECD and TMD, leading to channel opening. However, there are inconsistencies between these structures and previous functional work, particularly concerning charge selectivity, permeation, and single-channel conductance. One concern is that the pore of the previously reported open GlyR conformation (PDB_ID: 3JAE) is considerably wider than what has been seen in other anionic pLGIC structures, particularly at the selectivity filter[8]. Molecular dynamics (MD) simulations with this structure predict a channel conductance that is higher than that observed in electrophysiology studies and the structure is only partially blocked by the complete pore-blocking antagonist picrotoxin[12], suggesting that the pore opening may have been exaggerated. Moreover, in MD simulations performed in the absence of backbone restraints, the open structure has been shown to rapidly collapse at the Leu9′ region of the pore[13,14]. Potential causes for these discrepancies are that the previous structures were solved with a truncated ICD (consisting of 75–125 residues) to improve sample stability and in a detergent environment. The ICD, besides its role in trafficking, has also been implicated to alter single-channel conductance[15], modulation of gating[16], and desensitization[17]. In addition, a recent report shows that ICD truncation in GlyR leads to an increase in maximum single-channel open probability and a substantial increase in agonist efficacy[18]. Furthermore, lipids are essential components for membrane protein stability, and it therefore seems likely that both the ICD and lipid environment are important for maintaining the structural integrity of the channel during gating conformational cycles. Lipid nanodisc derived from membrane scaffolding protein (MSP) is one of the tools used to mimic the lipid bilayer environment[9,19]. We therefore determined high-resolution structures of the full-length GlyR in a membrane nanodisc

environment by single-particle cryo-electron microscopy (Cryo-EM) under conditions that stabilize various functional states. This allowed us to elucidate ligand-induced conformational changes that give a complete mechanistic description of gating.

## Results

**Cryo-EM structures of GlyR gating conformations.** The zebrafish GlyRα1 (Supplementary Table 1) exhibits robust macroscopic currents in response to application of glycine with an EC$_{50}$ of 100 μM with current amplitude saturating beyond 1 mM[20]. In the continued presence of glycine, the channels desensitize within minutes. Among the blockers of open-channel pore of anionic pLGICs, include picrotoxin (PTX), a plant alkaloid that occurs as an equimolar mixture of picrotin and picrotoxinin. While picrotoxinin is a more potent inhibitor in GABA$_A$Rs, both components are found to be of similar efficacy in GlyRs[21]. Upon application of PTX, GlyR currents are strongly inhibited with an IC$_{50}$ of 5–9 μM, and a near complete inhibition is achieved at concentrations >1 mM[22]. The block of GlyR currents is reversible, and previous studies have demonstrated that PTX block prevents the channel from desensitizing[23]. Our strategy to capture the GlyR in the resting, open, and desensitized conformations, was therefore to prepare samples of GlyR in the absence of a ligand, in the presence of both glycine and PTX, and in the presence of glycine, respectively (Supplementary Fig. 1). Ideally, one would hope to capture the channel in the open conformation in the absence of any perturbation (mutation, blocker, or potentiator). However, given the transient nature of the open conformation and the time requirements of cryo-EM sample preparation, extensive efforts toward this goal did not yield the desired outcome. Since reported mutations do not completely obliterate desensitization, we resorted to the use of PTX.

The full-length glycine α1 receptor (GlyR) gene from Zebrafish was codon-optimized, cloned into pFastBac1 plasmid, and expressed in *Spodoptera frugiperda* (Sf9) cells (Supplementary Table 2, Supplementary Fig. 2). The sample stability was improved by incorporating a lipid mixture consisting of soybean polar extract (asolectin) and cholesteryl hemisuccinate (CHS) during detergent solubilization and purification. The purified pentameric population of GlyR was then reconstituted into asolectin nanodiscs with the E3D1 MSP and the nanodisc samples were used for single-particle cryo-EM analysis (Supplementary Fig. 3). The samples were imaged in the absence of ligand (GlyR–Apo), in the presence of 5 mM glycine (GlyR–Gly), and in the presence of 5 mM glycine and 3 mM picrotoxin (GlyR–Gly/PTX). Three dimensional reconstructions with imposed C5 symmetry for the three samples led to density maps with nominal resolutions of 3.33 Å, 3.47 Å and 3.51 Å, respectively (Supplementary Fig. 4, Supplementary Table 3). In each case, the final reconstruction contained density for the entire ECD and TMD as well as a select region of the ICD, which were used for model building and refinement (Supplementary Fig. 5). The ICD region between Phe341 and Lys394, which is a part of the M3–M4 loop, is unstructured and appears as segmented density that is not conducive to model building. The density surrounding the TMD corresponds to the nanodisc belt comprising of helical segments of the MSP and lipid bilayer within the enclosure (Supplementary Fig. 3C). There are three sets of additional nonprotein densities in each subunit; one appears as an extension from Asn62 corresponding to N-glycans and the other two are lipid-like densities (in GlyR–Apo) in the vicinity of M4 (Supplementary Fig. 3C). The glycan density at Asn62 was also observed in previous structures[8,11] and corresponds to the single glycosylation site shown in GlyRα1. Mutations at this site have been reported to affect surface expression of functional GlyRs by preventing their exit from the endoplasmic reticulum[24]. The lipid-like densities at the extracellular end was modeled as phosphatidylcholine and the other at the cytosolic end

was partially modeled as a phospholipid (Supplementary Fig. 6). Interestingly, the density at the intracellular end is in a similar location to PIP2 density observed in a recent GABA$_A$R structure[9].

The overall architecture, consistent with the previous structures of GlyR and other pLGICs[8,11], consists of a symmetrical arrangement of the five subunits with dimensions of 125 Å in height and 80 Å in diameter. Each subunit consists of a twisted β-sheet of ten strands forming the ECD, four α-helical strands comprising the TMD, and a primarily unstructured region between the third and fourth TM helices forming the ICD (Supplementary Fig. 2B). The TM helices splay outward toward the extracellular end of the bilayer. The M4 helix extends out of the putative bilayer boundary (referred to as post-M4), by three turns of the helix and makes contact with the ECD. The ICD is the least conserved region among the pLGIC family and varies both in length as well as amino acid sequence. Based on the ICD sequence, one expects extensive structural variations between anionic and cationic pLGICs, consistent with the distinct roles of this domain in the two channel classes. In GlyR, the M3–M4 linker extends as a continuous stretch of α-helix from the C-terminal end of M3 up to four turns (referred to as post-M3). Beyond this, it is predicted to be an unstructured region that eventually terminates with a short α-helix (referred to as pre-M4) that extends into the M4 helix. In contrast, the ICD in cationic pLGICs consists of an unstructured loop (post-M3 loop), followed by an amphipathic MX helix that runs parallel to the membrane, an unstructured region that ends in a long α-helix MA which appears as a single continuous helix with M4[25]. In the resting conformation of cationic pLGICs, both the post-M3 loop and the MA helix occlude the ion permeation pathway. In 5-HT$_3$AR, ligand-binding elicits large conformational changes in this region to create a lateral portal for ion exit[6,7]. In the absence of extensive structured motifs, it remains unknown if the ICD in the anionic pLGIC exerts such a barrier to ion permeation.

The GlyR–Apo, GlyR–Gly/PTX, and GlyR–Gly structures reveal distinct conformational states of the ion permeation pathway, which originates at the ECD, extends through the TMD, and terminates shortly within the ICD (Fig. 1a). Within the TMD, the pore is lined by the M2 helix from each subunit in an arrangement that appears more cylindrical in GlyR–Apo and becomes progressively funnel shaped, tapering toward the intracellular end, in GlyR–Gly/PTX and GlyR–Gly. In the GlyR–Apo, the pore is relatively narrow and constricted prominently at Leu9′ (pore radius ~1.4 Å) and Thr13′ (pore radius ~2.2 Å). The pore dimensions at positions Ala20′ and Pro-2′ are also below the Born radius for the solvated chloride ion, which is 2.26 Å, (the Pauling radius for chloride ion is 1.81 Å) and are therefore likely barriers to ion permeation (Fig. 1b). The M2 helices show partial unwinding between Gly17′ and Ala20′, a feature that is consistent with the dynamic behavior of this region previously noted in nuclear magnetic resonance studies of the isolated GlyR TMD[26]. In the GlyR–Gly/PTX structure, the M2 helices are oriented outward with the Leu9′ rotated away from the central axis (Fig. 1c). The pore radius profile (created with the PTX ligand removed from the pore) shows an expanded ion permeation pathway at each of the constriction points seen in GlyR–Apo. In GlyR–Gly, the pore is similar in radii to that of the GlyR–Gly/PTX but reveals a notable constriction (pore radius ~1.8 Å) at the level of Pro-2′ due to slight bending of the intracellular end of M2. The Pro-2′ position is part of the charge selectivity filter in anionic channels

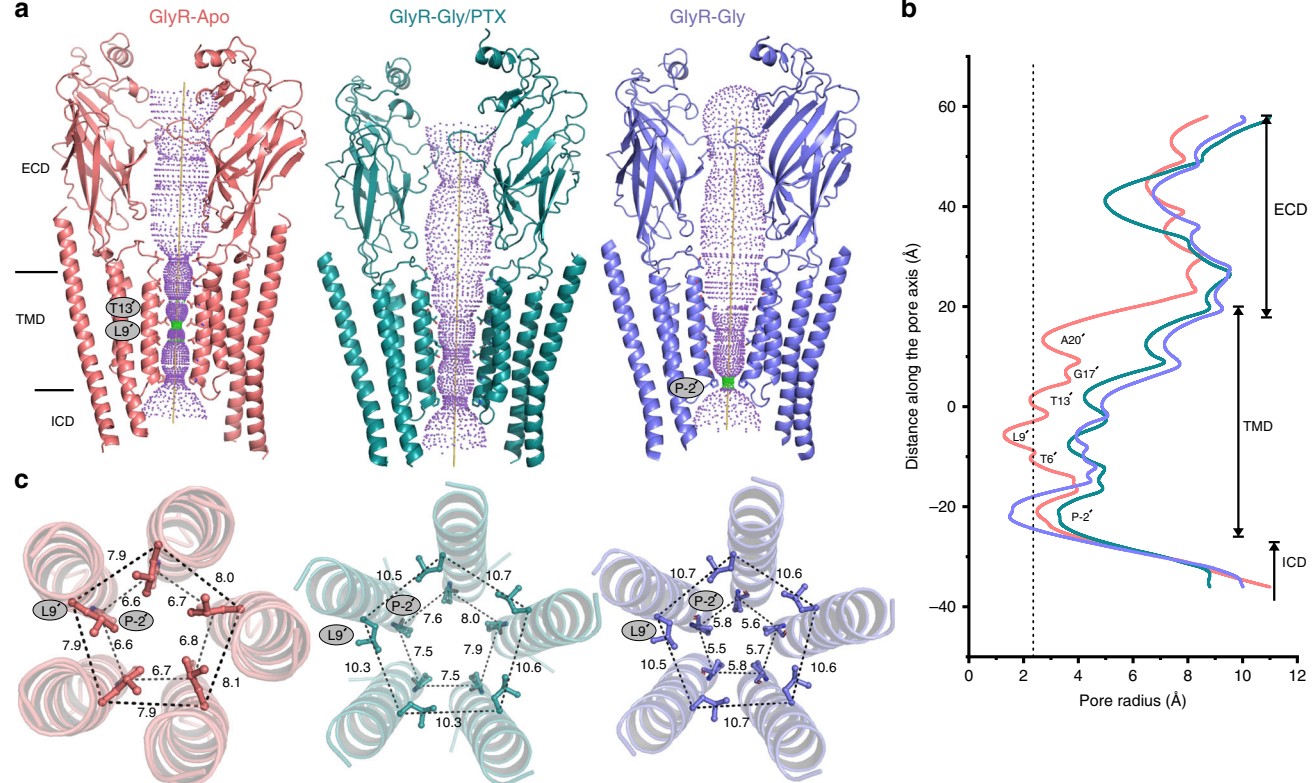

**Fig. 1 Cryo-EM structures of full-length GlyR in multiple conformational states. a** Ion permeation pathway generated with HOLE[60] for GlyR–Apo (salmon red), GlyR–Gly/PTX (deep teal) and GlyR–Gly (slate blue). For clarity, the cartoon representation of only two non-adjacent subunits are shown. Green and purple spheres define radii of 1.8–3.3 Å and >3.3 Å, respectively. The residues located at various pore constrictions are shown as sticks. **b** The pore radius is plotted as a function of distance along the pore axis. The dotted line indicates the approximate radius of a hydrated chloride ion, which is estimated at 2.26 Å. **c** A view of M2 helices from the extracellular end for the three GlyR conformations. Positions Leu9′ and Pro-2′ are shown in ball-and-stick representation and the corresponding distances between Cα are given in Å.

comprising of a conserved stretch (Pro-2′, Ala-1′, and Arg0′)[27] and mutations in this stretch are associated with hyperekplexia[28]. While Pro-2′ and Ala-1′ face the lumen of the pore, Arg0′ faces away. Previous studies have shown that both the charge and the side-chain conformation of Arg0′ are determinants of anion selectivity[29].

In addition to the physical dimensions of the pore, the hydrophobicity of the pore-lining residues plays a key role in dictating the conductance state of the channel. In some cases, a pore that is not narrow enough for steric occlusion may still impede ion permeation due to the hydrophobicity of the pore-lining residues that may disfavor water molecules, causing local dewetting that leads to a higher energetic barrier for water and ion permeation[30]. We assessed the combined effect of both the pore radii and hydrophobicity on pore dewetting and presence of hydrophobic gates using a previously developed simulation-free heuristic model that was derived using machine learning from MD simulations of nearly 200 ion channel structures[31]. A pore surface profile based on the radii of the permeation pathway and hydrophobicity of the pore-lining amino acid side chains was estimated by CHAP[32] (Supplementary Fig. 7A). On the plot of the local pore radii at each pore-lining residue against its side-chain hydrophobicity, the dotted line marks the divide of the hydrophobicity-radii landscape into regions of high and low likelihood of pore wetting (Supplementary

Fig. 7B). The sum of shortest distances ($\Sigma d$) from positions falling in the low-likelihood region to the line is used as a heuristic score which allows a prediction for the likelihood of the conformation corresponding to a dewetted and nonconductive state. A cut-off value of $\Sigma d > 0.55$ is used for predicting a nonconductive conformation. In the GlyR–Apo, several residues (Pro-2′, Ile5′, Leu9′, and Thr13′) fall below the dotted line, leading to an overall heuristic score of $\Sigma d = 1.11$. In the GlyR–Gly/PTX, there are no residues below the line and the score is 0.0. On the other hand, in the GlyR–Gly structure, Pro-2′ is well below the line with Val1′ lying close and the score is 0.80. These findings suggest that the pore in Gly–Gly/PTX structure is likely to be open, whilst those in the GlyR–Apo and GlyR–Gly structures are closed. The electrostatic map shows the positively charged cluster at the intracellular end of M2 and the regions of the ICD close to the TMD interface (Supplementary Fig. 7C). The Arg0′ that is implicated to be the selectivity filter in anionic pLGIC lies on the non-pore facing side of M2 at the interface of the TMD and ICD.

The GlyR–Gly and GlyR–Gly/PTX structures were solved in the presence of 5 mM glycine and the 3D reconstructions reveal an additional small density in the canonical neurotransmitter-binding pocket wedged at the subunit interface in the ECD (Fig. 2a). An unambiguous assignment of a small ligand such as

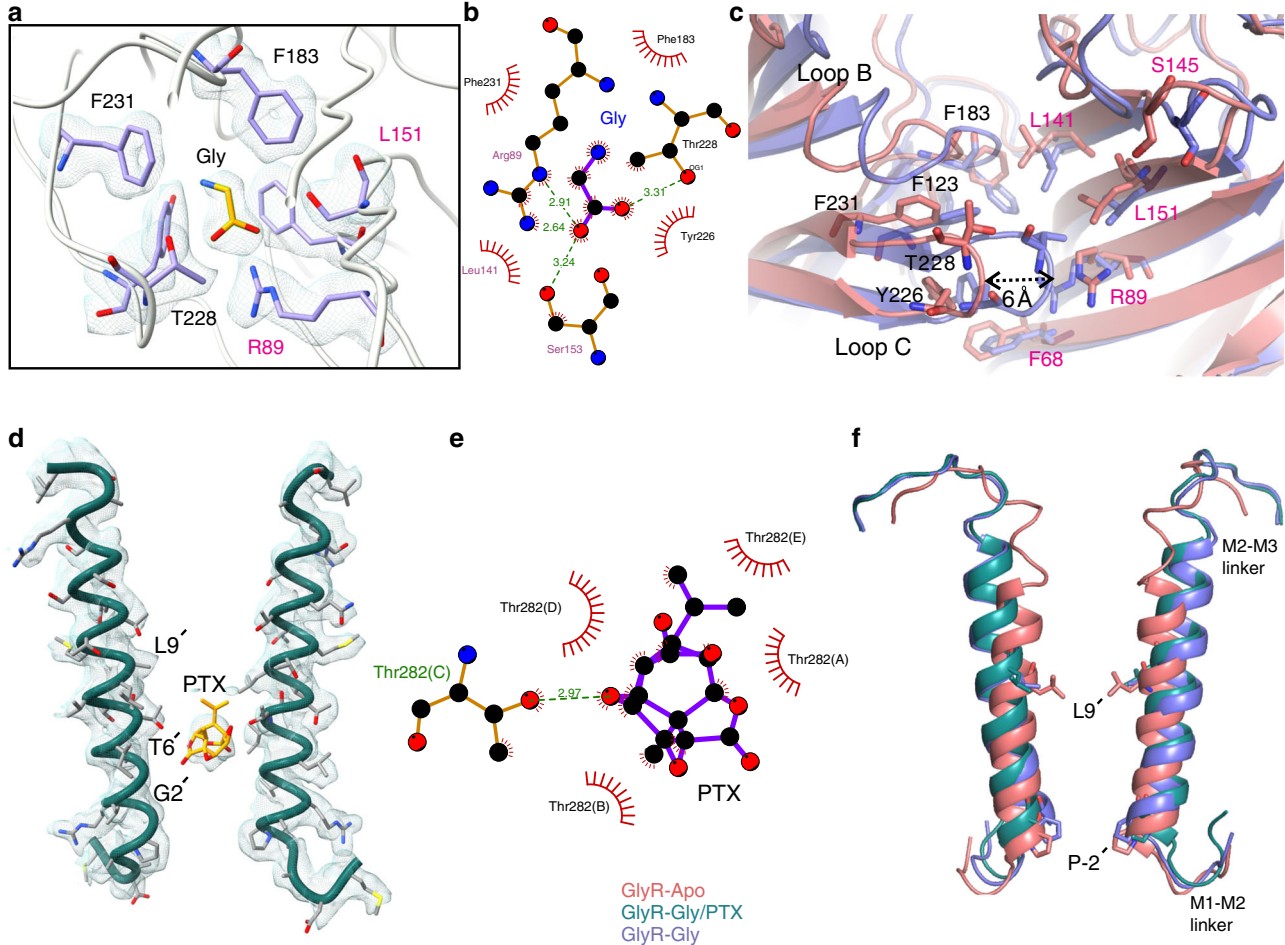

**Fig. 2 Conformational changes underlying GlyR gating. a** Cryo-EM density segments for neurotransmitter binding site residues and glycine ligand as seen in the GlyR–Gly structure. Residues in the principal subunit and complementary subunit are indicated in black and magenta, respectively. **b** LigPlot analysis of glycine orientation in the binding pocket and residues within 4 Å distance are displayed in red[70]. **c** Comparison of the neurotransmitter binding site for the GlyR–Apo and GlyR–Gly conformations. The residues that are involved in neurotransmitter binding are shown in sticks. **d** Cryo-EM map showing the density for M2 and PTX bound in the pore of GlyR–Gly/PTX. The interacting residues are shown in stick representation. For clarity, M2 for only two diagonal subunits are shown. **e** LigPlot analysis of PTX and the interacting residues. **f** A close-up of the M2 conformations is shown upon aligning the three GlyR conformations. Positions Leu9′ and Pro-2′ are shown as sticks.

glycine is difficult at this resolution, however, since the density was not seen in GlyR–Apo, and the location matched with the glycine density in the α3-GlyR crystal structure[33], we modeled the glycine ligand accordingly. The neurotransmitter binding pocket is lined by several conserved aromatic residues and bulky side-chains, from both the principal (Phe123, Phe183, Tyr226, Thr228, and Phe231) and complementary subunits (Phe68, Arg89, Leu141, Ser145, and Leu151). LigPlot analysis shows interaction of glycine with several of these residues (Fig. 2b). Mutational studies have shown that a significant impact on agonist affinity ensues upon perturbation of this pocket[34]. An overlay of GlyR–Apo and GlyR–Gly shows substantial conformational changes that lead to a rearrangement of the residues in the binding pocket (Fig. 2c). The major motion includes an inward movement of Loop C (carrying Tyr226, Thr228, and Phe231) and Loop B (Phe 183). Upon glycine binding, the tip of Loop C (measured at Cα of Thr228) is displaced by 6.2 Å toward the neurotransmitter binding pocket thereby shrinking the pocket and potentially occluding water molecules. The inward or "capping" motion of Loop C is well-described in pLGIC literature and is associated with agonist-binding[35] and is also observed in the recent structures of the unliganded and agonist-bound conformations[7,8,36].

In the presence of glycine, PTX reversibly blocks GlyR currents (Supplementary Fig. 1). In the GlyR–Gly/PTX reconstruction, in addition to the glycine density at the neurotransmitter binding pocket, an additional density was seen in the pore at the mid-level of M2 nestled between Gly2′ and Leu9′ (Fig. 2d). This density was not observed in the other two conditions and likely corresponds to a PTX molecule. A recently solved structure of GABA_AR also contains a PTX molecule at a similar location and was used to guide the orientation of the molecule[37]. The hydrophobic isoprenyl end of PTX is oriented toward the Leu9′ and the hydrophilic end closer to the Thr6′ (Fig. 2e). Mutational studies have shown that perturbations at these positions affect PTX binding in anionic pLGICs[38]. It is noteworthy that mutations at the Thr6′ position can abolish PTX sensitivity in α1GlyR and enhance PTX sensitivity in PTX-insensitive α1β GlyR[38]. An inspection of the M2 helices upon global alignment of all three conformations show that the binding site for PTX does not physically overlap with the region undergoing constriction in the GlyR–Gly (Pro-2′). An alignment of GlyR–Gly and GlyR–Gly/PTX shows that the two structures differ primarily at the level of Pro-2′, the desensitization gate (Fig. 2f, Supplementary Fig. 8, green box). It is interesting to note that PTX does not induce additional conformational changes at the binding site (Supplementary Fig. 8, gray box). Therefore, it appears that PTX exerts an allosteric effect on the intracellular gate Pro-2′ and prevents it from closing in the presence of glycine. In addition, we note positional differences in PTX in GlyR–Gly/PTX and GABA_AR (α1β3γ2) structures compared with GluCl (Supplementary Fig. 9) where PTX interacts with Thr2′ instead. We think this variation may arise from the nature of the side chain at the 2′ position (Gly in homomeric α1GlyR, Val/Ala/Ser in α1β3γ2 GABA_AR, or a Thr in GluCl) or from slightly different mechanisms of PTX action as is also exemplified by use-dependent nature of PTX block in GABA_AR and GluCl but not in the case of GlyR[39,40].

When compared to GlyR–Apo, the glycine-bound structures undergo an anti-clockwise twist of the ECD around the pore axis (when viewed from the extracellular end) (Supplementary Movie 1). The ECD twist is accompanied by an inward motion of Loop C described above (Supplementary Fig. 10). In addition, there is major repositioning of the interfacial loops, particularly the β1–β2 loop and Cys-loop (β6–β7) from the principal subunit and Loop F (β8–β9) on the complementary subunit (Fig. 3a). The TMD helices are rotated clockwise and tilted outward in the

GlyR–Gly/PTX and GlyR–Gly structures compared with GlyR–Apo, leading to an iris-like opening of the TMD (Supplementary Fig. 10, Fig. 3b). In addition to the M2 helices moving outward to open the pore, the M2–M3 linker is also pulled away from the pore axis. This twisting motion is similar to the previously observed changes in GlyR and other pLGICs[6–8]. The described motions have a substantial effect on interactions between subunits at both the ECD and TMD interfaces as well as intra-subunit interactions both in the ECD and TMD (Fig. 3c). In GlyR–Apo, the Cys-Loop, β1–β2 linker, and Loop F from the ECD are in close contact with the M2–M3 linker, pre-M1, and post-M4. In addition, there are several interactions between M1, M3, and M4 helices between two adjacent subunits. In the glycine-bound conformations, these interactions are reduced due to the ECD rotation and the TMD expansion at the extracellular end (Fig. 3c). The overall buried surface area (regions inaccessible to water) of the pentameric assembly in GlyR–Apo is 36,530 Å² and reduces to 26,210 Å² and 26,720 Å² in GlyR–Gly/PTX and GlyR–Gly, respectively. The interaction network at the domain interfaces are crucial signal transduction elements and, unsurprisingly[41], mutational perturbations at several of these positions (marked in green, Fig. 3c) are reported to significantly alter function, and are associated with disease pathology including hyperekplexia[2].

Glycine-induced conformational changes within the TMD leads to expansion of the TM helical bundle (Fig. 4a). In GlyR–Apo, M4 helices are oriented closer to the rest of the TM helices within the same subunit. The post-M4 region extends three turns above M4, and extends above the putative membrane and is positioned close to the pre-M1 region and the β8–β9 strand of the ECD (Supplementary Fig. 2B). These structural elements are implicated in relaying signals from the ECD to the TMD[42]. In GlyR–Gly, these regions are placed farther away. At the intracellular end, the M4 and the pre-M4 helices (intracellular extension) are bent in the vicinity of Pro419 and positioned in close proximity to a lipid-like density (Supplementary Fig. 6). In the GlyR–Gly structure, the M4 and pre-M4 helices are further straightened (Fig. 4b), and there is no obvious lipid-like density at this position. The movement of post-M3 helices is in the same direction as the pre-M4, and is potentially coupled by the unstructured loop that is currently not modeled in our structures. Among other related movements at the intracellular end is the movement of the M1–M2 linker (Fig. 4b). In the GlyR–Gly structure, the M1–M2 linker is moved away from the pore axis as a consequence of (or to facilitate) the pinching of M2 helices at Pro-2′. In agreement, both the length and the sequence of the M1–M2 linker affect GlyR gating and desensitization, presumably by affecting the positioning of the selectivity filter region (Pro-2′, Ala-1′, and Arg0′)[17,43]. Another significant consequence of structural changes at the TMD and ICD are the appearance of intra-subunit and inter-subunit cavities at both the extracellular and intracellular ends of the TMD (Fig. 4c). GlyRs are targets to numerous endogenous and exogenous ligands including lipids, neurosteroids, cannabinoids, alcohols, and anesthetics that bind within the TMD cavities. A change in the volume and polarity of these cavities implicates a role in state-dependent effects of the modulators on channel function[44]. Several residues lining these pockets (Fig. 4c) are shown to modulate binding of these allosteric ligands and their accessibility is increased in glycine-bound conformation compared to the resting state[45].

**Allosteric modulation of GlyR gating.** Among the several types of allosteric modulators of GlyR that bind within the TMD cavities, we studied a well-described potentiator, ivermectin (IVM), which is known to enhance Gly sensitivity and increase open

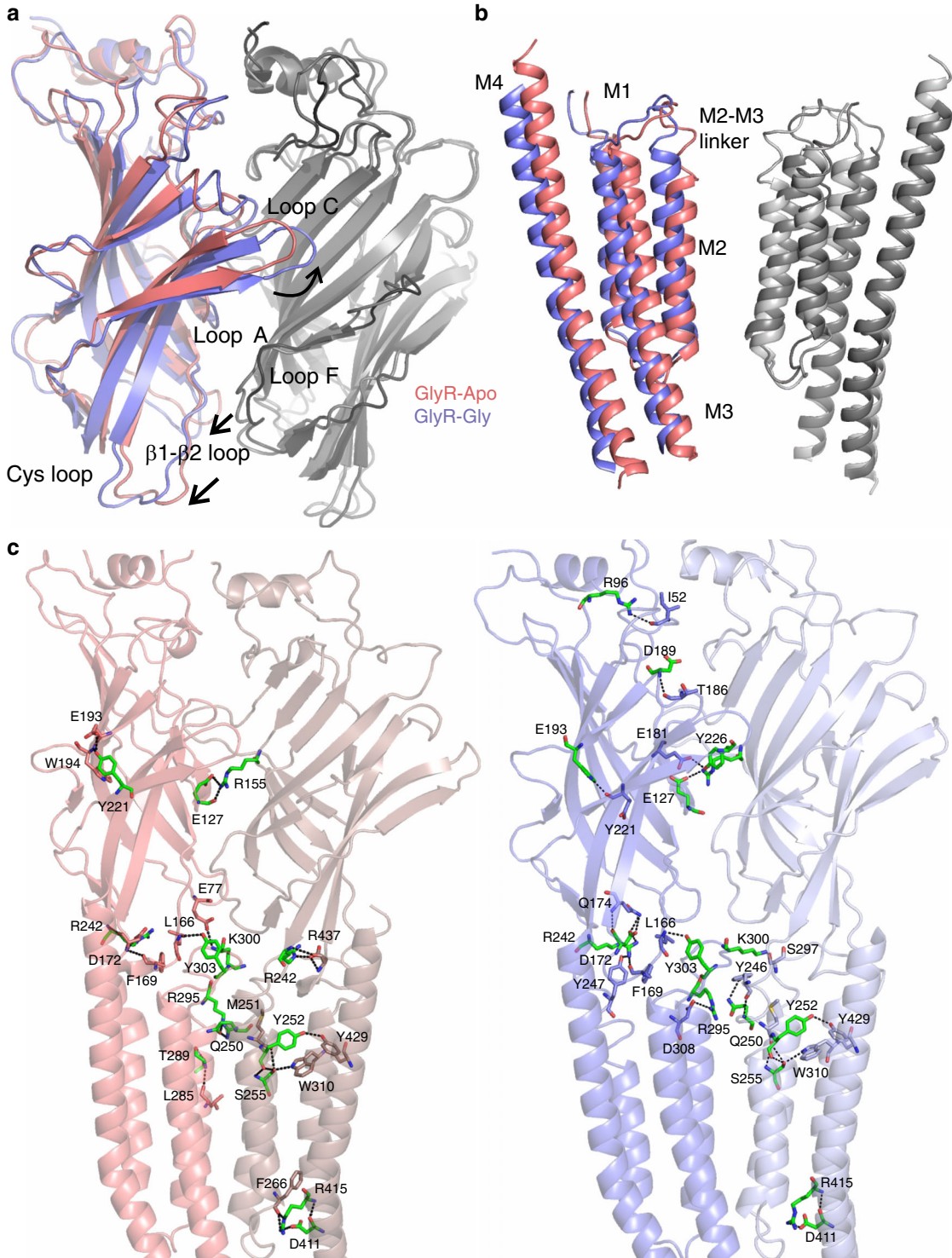

**Fig. 3 Global conformational changes and altered inter-domain interactions during GlyR gating. a** A side-view of the ECD interface in GlyR–Apo and GlyR–Gly. The principal subunits are colored (GlyR–Apo: salmon red and GlyR–Gly: slate blue) while the complementary subunits are shown in shades of gray. **b** A side view of the TMD in GlyR–Apo and GlyR–Gly conformations. **c** Interactions at the subunit- and domain-interfaces between ECD–ECD, ECD–TMD, and TMD–TMD in GlyR–Apo (left) and GlyR–Gly (right). The residues involved in the interactions are shown in sticks and interactions are highlighted by black dashes. Mutations at positions shown in green are associated with hyperekplexia[2].

probability (Po) at lower glycine concentrations. There are several IVM-bound structures (both from cryo-EM and X-ray crystallography) for GlyR and GluCl that have provided initial views of the IVM-binding site[8,33,46]. We solved the structures of full-length GlyR in nanodisc in the presence of varying concentrations

of glycine and IVM (1 mM Gly/0.5 μM IVM and 0.1 mM Gly/30 μM IVM) (Fig. 5). The first condition led to two states (Gly-IVM-1 and Gly-IVM-2) and the second condition led to one state (Gly-IVM-3). These structures are in the 3.0–3.3 Å resolution range with excellent map quality (Supplementary Figs. 11 and 12;

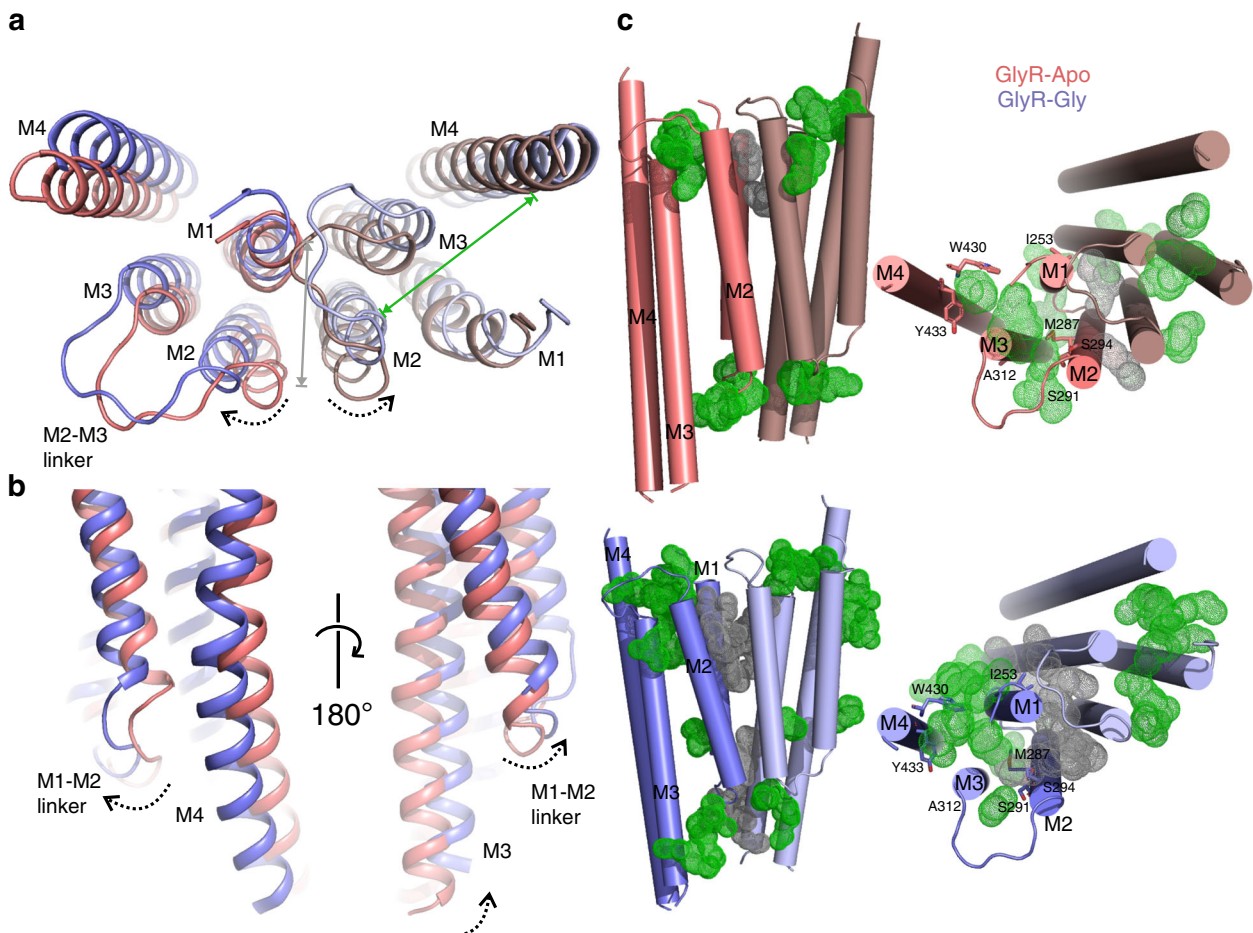

**Fig. 4 Conformational changes in the M4 helix and the effects on internal cavities. a** An overlay of GlyR–Apo and GlyR–Gly. The TMD viewed from the top reveals repositioning of TM helices. The intra-subunit and inter-subunit cavities are indicated by green and gray arrows, respectively. **b** Extent of conformational change in the ICD formed by the pre-M4 region, the M1–M2 linker, and the post-M3 region. The direction of movement is highlighted by black arrows. **c** A comparison of the inter-subunit (gray dots) and intra-subunit (green dots) cavities in GlyR–Apo (top) and GlyR–Gly (bottom) predicted using F pocket algorithm[61]. The cavities predicted on the surface were removed for clarity. The right and left panels are side and top views, respectively. The residues shown in sticks on the left panel are implicated in binding allosteric ligands[45].

Supplementary Table 4). Each of these three states reveal clear densities for both glycine and IVM (Supplementary Fig. 13). The IVM molecule binds at the subunit interface between M3 (from the principal subunit) and M1 (from the complementary subunit) and makes polar contact with Arg295 in M2 (from the principal subunit). The binding pocket is lined by several hydrophobic residues (Ile249, Ile253, Pro254, and Leu257 in M1, Val304, Ala312, Leu315 in M3) with TM helices. The orientation of IVM is as previously observed in other GlyR–IVM structures.

The overall conformation of the three structures are very similar to GlyR–Gly. At the level of M2, the channel is open at Leu9′ and constricted at Pro-2′, broadly reminiscent of GlyR–Gly (Fig. 5). A closer inspection of the six GlyR structures reveals that the Pro-2′ side chains are positioned clockwise going from Apo, Gly/PTX, to Gly conformations. The Pro-2′ sidechain in each of the three IVM structures lies between that of the Gly/PTX and Gly, suggesting that these conformations may represent intermediate states between the open and desensitized states (Fig. 5d).

We compared the full-length GlyR states with the previously solved GlyR cryo-EM conformations by aligning GlyR–Apo with StryR (strychnine-bound), GlyR–Gly/PTX with Gly (glycine-bound open), and GlyR–Gly and GlyR–Gly/IVM-3 with IVM (glycine/ivermectin-bound)[8]. While the structures overlap well (Supplementary Fig. 14), the most notable differences for each

alignment pair are in the TMD and the domain interfaces. The buried surface area of StryR is 27,392 Å² with greater numbers of inter-domain interactions compared with that of 36,530 Å² in GlyR–Apo. The structural elements comprising the ICD adopt distinct conformation, namely the M1–M2 linker and the post-M3 and pre-M4 helices. In addition, in comparison with the GlyR–Gly/PTX, the glycine-bound open state is much wider, particularly at the intracellular end where the charge selectivity filter of the channel resides (Arg0′). We believe that some of these differences may arise from truncation of the construct and the presence of a detergent environment. A comparison of the apo, open, and desensitized conformations of GlyRs with the corresponding states from other anionic pLGIC (Supplementary Fig. 15), shows an overall conservation of the direction of agonist-induced movements in the ECD and TMD. However, the structures differ in the extent of repositioning of the loops and TM helices.

**Assessing the conductance state of GlyR structures.** By evaluating the hydrophobicity and radius of the transmembrane pore, the constrictions at Leu9′ and Pro-2′ were predicted to form energetic barriers to varying extents in each of the GlyR conformational states. To assess the behavior of water molecules and

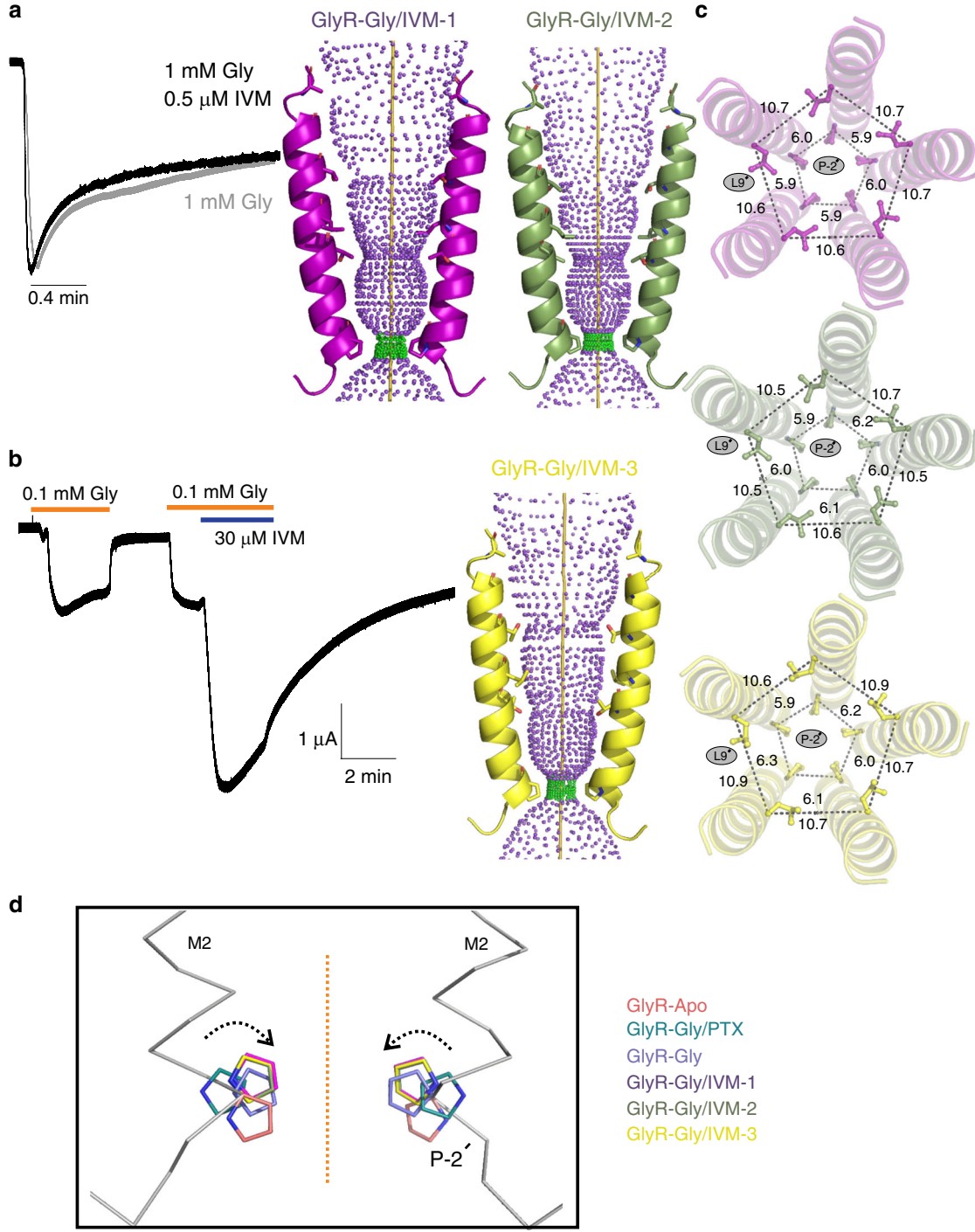

**Fig. 5 Analysis of GlyR–Gly/IVM structures. a** TEVC recording of 1 mM glycine-induced currents in the presence and absence of 0.5 μM IVM (left). Representative trace from multiple independent oocyte recordings ($n = 5$). The traces are normalized to the peak current to show the current decay. Ion permeation pathway for GlyR–Gly/IVM-1 GlyR–Gly/IVM-2 (right). Green and purple spheres define radii of 1.8–3.3 Å and >3.3 Å, respectively. The residues located at various pore constrictions are shown as sticks. **b** Current recording for 0.1 mM glycine and 0.1 mM glycine and 30 μM IVM (left). Representative trace from multiple independent oocyte recordings ($n = 3$). The pore profile for GlyR–Gly/IVM-3 structure. **c** A view of M2 helices from the extracellular end Positions Leu9′ and Pro-2′ and the corresponding distances between Cα in Å. **d** Alignment of GlyR structures showing the conformational differences at the level of Pro-2′. Dotted arrows indicate the direction of rotation going from the GlyR-Apo (closed) to the GlyR–Gly (desensitized) state.

ions for the GlyR–Apo, GlyR–Gly/PTX, and GlyR–Gly conformations in a membrane environment, MD simulations were carried out upon embedding the GlyR structures within a 1-palmitoyl-2-oleoyl-*sn*-glycero-3-phosphocholine lipid bilayer (Fig. 6). The picrotoxin molecule in the pore of GlyR–Gly/PTX structure was removed prior to equilibration in the membrane.

Positional restraints were placed on the protein backbone during these simulations, preserving the overall experimentally determined conformational state while permitting rotameric flexibility in amino acid side chains. Water molecules (using the TIP4P/2005 model) and ~150 mM NaCl were included on either side of the bilayer. During simulations, the side-chain movements of the

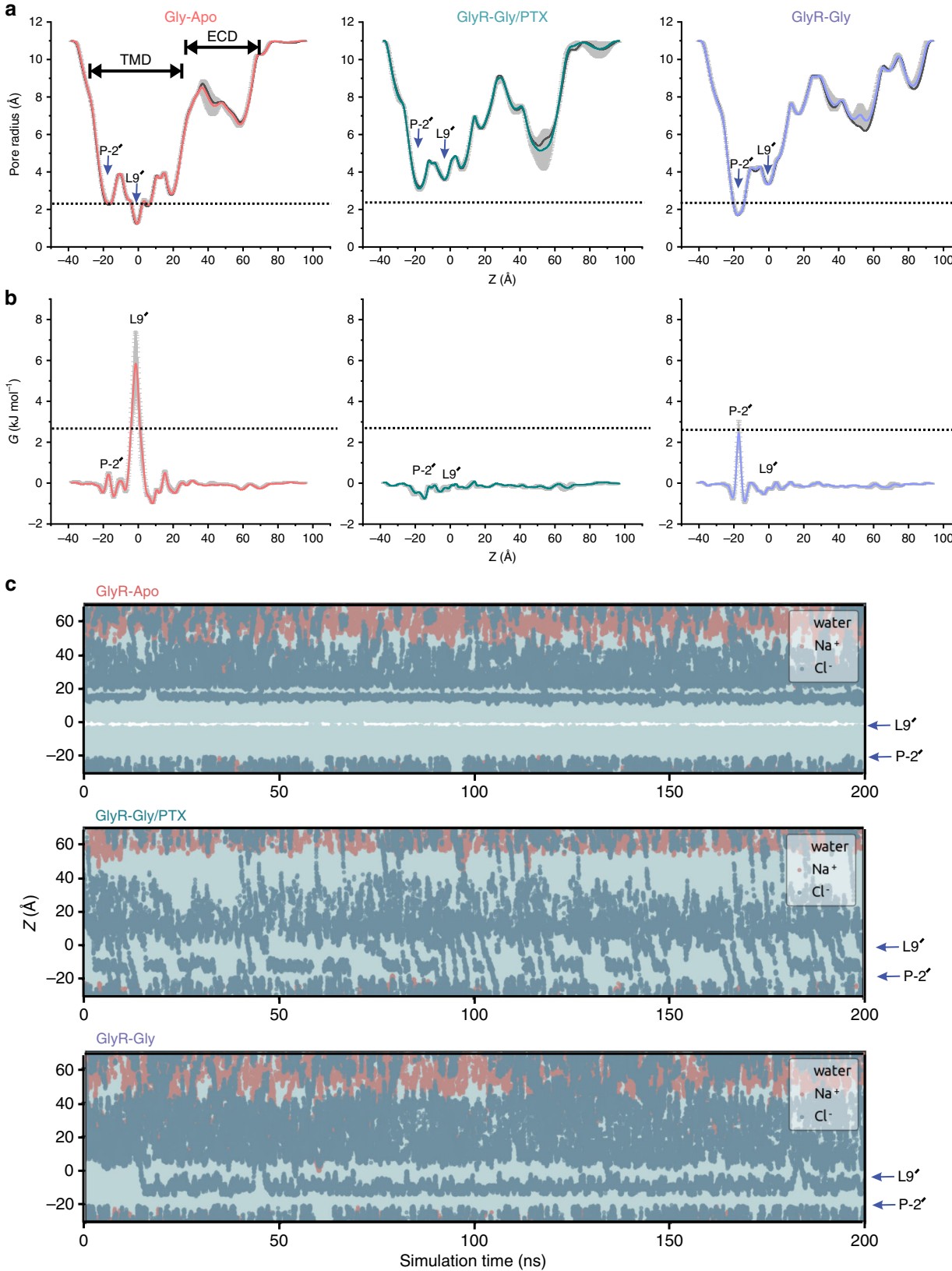

pore-lining residues caused fluctuations in the pore-radius, but there were no major changes to the pore profile (Fig. 6a). An analysis of simulated water density along the pore axis suggests that in GlyR–Apo the pore is dewetted (i.e. devoid of water molecules) at Leu9′ (at ~0 Å), conferring closure to water (with an

energetic barrier of ~5 kJ mol$^{-1}$) (Fig. 6b). This barrier disappears in both the GlyR–Gly/PTX and GlyR–Gly structures. However, while the pore is hydrated in GlyR–Gly, a small energetic peak (~2.5 kJ mol$^{-1}$) appears at the Pro-2′ position (~−20 Å), which corresponds to a maximum free energy value of ca. 1 RT (i.e.

**Fig. 6 Molecular dynamics simulations of GlyR conformations. a** Mean pore radius profiles and standard deviations averaged across three independent 30 ns equilibrium simulations for GlyR–Apo (left), GlyR–Gly/PTX (middle), and GlyR–Gly (right) along the central pore axis. The final 20 ns of each 30 ns simulation trajectory was used to evaluate these profiles. The one-standard-deviation range between calculations ($n = 3$, independent repeats of 30 ns simulations) are shown as a gray band and the thick line is the mean between the triplicate 30 ns simulations. Major constriction sites are indicated and the dotted line denotes the radius of hydrated chloride ion. The black traces are the pore radius profile calculated from the cryo-EM structures. **b** Corresponding mean water free energy profiles and standard deviations. Peaks in free energy profiles are highlighted. **c** Trajectories along the pore ($z$)-axis of water molecules and chloride ion coordinates within 5 Å of the channel axis inside the pore, in the presence of a +500 mV transmembrane potential difference (i.e., with the cytoplasmic side having a positive potential). One of five independent 200 ns replicates is shown for each structure. During these and the preceding simulations, positional restraints were placed on the protein backbone, in order to preserve the experimental conformational state while permitting rotameric flexibility in amino acid side chains. The energetic barriers due to the ring of Leu9′ and Pro-2′ are at $z$ ~0 and −20 Å, respectively.

comparable to that of thermal fluctuations). While water permeation across the pore is essential for ion permeation, it is not necessarily sufficient. To assess ion conductance for the three states, in a second set of simulations, a transmembrane potential of +500 mV (i.e. positive at the cytoplasmic side) was applied at 500 mM NaCl concentration. The elevated concentration and voltage were used to increase the likelihood of observing permeant ions in microsecond simulation times (in separate control simulations we have checked the linear dependence of simulated conductance on NaCl concentration over the range 0–500 mM). As expected, no ion permeation events were observed for GlyR–Apo, even at this elevated potential and concentration. Interestingly, while water permeates through GlyR–Gly, no chloride ions passed through the channel during 1 μs of total simulation time (Fig. 6c). In contrast, 173 chloride ion permeation events (defined as ions passing from end to end of the TMD of the intact receptor used in the simulation) were observed over the course of 1 μs (= 5 × 200 ns) of total simulation trajectory of the GlyR–Gly/PTX structure (Supplementary Movie 2). This corresponds to an estimated single-channel conductance of ~20 pS (corrected to the value at 0.15 M NaCl) for chloride. The conductance is smaller than experimental measurements from single-channel experiments, which showed a conductance of 80–88 pS[47]. We believe that some of the differences in the conductance estimate may be due to the missing residues in the ICD that may exert an effect on the single-channel conductance through electrostatic mechanisms[15], but they may also reflect limitations of the computational methodology. It is noteworthy that sodium ions do not pass past the selectivity filter region comprising of Arg0′ (~−20 Å), confirming the anionic selectivity for GlyR. Our computational assessment of the three GlyR conformations thus indicate that the GlyR–Gly/PTX structure should be assigned to an open state, whereas the GlyR–Apo and GlyR–Gly structures are nonconductive.

Several previous simulation studies have noted the instability of the pLGIC conformations in the absence of protein backbone restraints[12–14]. While the fluctuations in the pore radii are understandably larger, the defining pore features of the three conformations were stably maintained, namely, the constrictions and energetic barriers to water at Leu9′ and Pro-2′ in GlyR–Apo and at Pro-2′ in GlyR–Gly. In addition, unrestrained simulations (3 × 200 ns) indicate that the GlyR–Gly/PTX is stable, remaining open and presenting no free energy barrier to water. Overall, based on the pore-profile features of GlyR–Gly/PTX and GlyR–Gly which show that there are no additional effects of PTX binding besides changes at the P-2′ position, and the MD simulations, which confirm that the pore conformation of GlyR–Gly/PTX has close to the expected selectivity and single-channel conductance, we believe that this structure may be close to the physiological open conformation.

To access the conductance states of the IVM structures, we carried out MD simulations as described above. Under these conditions, no ion permeation event was observed at Pro-2′ (on a

200 ns timescale) suggesting that the IVM-bound structures are nonconducting (or at most have a reduced single channel conductance compared to the Gly/PTX) (Supplementary Fig. 16). Interestingly, none of three structures reveals a sizable energetic barrier to water permeation at Pro-2′, which is in contrast to that of GlyR–Gly. These findings further point toward the idea that IVM-bound structures may represent intermediate conformations, and subtle changes in Pro-2′ side-chain orientation underlie transitions between open and desensitized states.

## Discussion

Here, we present a set of structures that reveal the complete range of agonist-induced conformational changes in the full-length GlyR reconstituted in a lipid nanodisc environment. At the heart of pLGIC gating mechanisms is the signal transduction machinery that communicates agonist-binding events to the channel pore leading to channel opening. We show that glycine induces a global conformational change that encompasses the ECD, TMD, and the structured regions of the ICD. In the resting conformation, the GlyR has tightly coupled subunit and domain interfaces that weaken in response to glycine binding, leading to a relaxed open and desensitized conformations. MD simulations were used to assess the energy landscape of pore hydration and ion permeation, which allowed us to annotate these structures as resting, open, and desensitized conformations. The single-channel conductance and ion selectivity predicted for the open conformation are in agreement with the wealth of previous functional studies. Overall, these studies set the stage for a detailed characterization of the functional modulation of this clinically important class of channels.

## Methods

**Two-electrode voltage-clamp (TEVC) electrophysiology.** The gene encoding zebrafish GlyRα1 (purchased from GenScript) was inserted into *pTLN* vector for expression in *Xenopus laevis* (Supplementary Table 1). To linearize the DNA, the plasmid was incubated with *Mlu1* restriction enzyme at 37 °C overnight. The mRNA was synthesized from linearized DNA using the mMessage mMachine kit (Ambion) per instructions in the manufacturer's manual. The RNA was then purified with RNAeasy kit (Qiagen). About 2–10 ng of mRNA was injected into *X. laevis* oocytes (stages V–VI) and experiments were performed 2–3 days after injection. For control experiments to verify that no endogenous currents were present, oocytes were injected with the same volume of water. Dr. W.F. Boron kindly provided oocytes used in this study. Female *X. laevis* were purchased from Nasco. We complied with all relevant ethical regulations for animal testing and research. Animal experimental procedures were approved by Institutional Animal Care and Use Committee (IACUC) of Case Western Reserve University. Oocytes were maintained at 18 °C in OR3 medium (GIBCO-BRL Leibovitz medium containing glutamate, and 500 units each of penicillin and streptomycin, with pH adjusted to 7.5 and osmolarity to 197 mOsm). TEVC experiments were performed on a Warner Instruments Oocyte Clamp OC-725. Currents were sampled and digitized at 500 Hz with a Digidata 1332A, and analyzed by Clampfit 10.2 (Molecular Devices). Oocytes were clamped at a holding potential of −60 mV and solutions were changed using a syringe pump perfusion system flowing at a rate of 6 ml/min. The electrophysiological solutions consisted of (in mM) 96 NaCl, 2 KCl, 1.8 CaCl₂, 1 MgCl₂, and 5 HEPES (pH 7.4, osmolarity adjusted to 195 mOsM). Chemical reagents (glycine and

picrotoxin) were purchased from Sigma-Aldrich. Current traces were plotted using Origin Version b9.5.0.193.

**Full-length GlyR cloning and transfection.** The zebrafish GlyRα1 shares 92% amino acid similarity with the human GlyRα1. Codon-optimized zebrafish GlyRα1 (NCBI Reference Sequence: NP_571477) was purchased from GenScript (Supplementary Table 2). The sequence included the full-length GlyRα1 gene (referred to in the text as GlyR) followed by a thrombin sequence (LVPRGS) and a C-terminal octa-His tag. The gene was subcloned into the pFastBac1 vector for expression in *Spodoptera frugiperda* (*Sf9*) cells. The *Sf9* cells (purchased from Invitrogen) were cultured in Sf-900™ II SFM medium (Gibco®) without antibiotics and incubated at 28 °C without CO$_2$ exchange. Transfection of sub-confluent cells was carried out with recombinant GlyR bacmid DNA using Cellfectin II transfection reagent (Invitrogen) according to manufacturer's instructions. Cell culture supernatants were collected at 5 days post-transfection and centrifuged at 1000×g for 5 min to remove cell debris to obtain progeny 1 (P1) recombinant baculovirus. *Sf9* cells were infected with P1 virus stock to produce P2 virus, which were then used for subsequent infection to produce P3 viruses and so on till the P6 generation. The P6 virus was used for recombinant protein production based on expression levels and eventually quality of the purified sample.

**GlyR purification and nanodisc reconstitution.** Approximately, 2 × 10$^6$ per ml *Sf9* cells were infected with P6 recombinant viruses. After 48 h post infection, the cells were harvested and centrifuged at 8000g for 20 min at 4 °C to separate the supernatant from the cell pellet. The cell pellet was resuspended in dilution buffer (20 mM Tris-HCl, pH 7.5, 36.5 mM sucrose) supplemented with 1% protease inhibitor cocktail (Sigma-Aldrich). Cells were disrupted by sonication on ice and nonlysed cells were removed by centrifugation at 3000×g for 15 min. The supernatant was subjected to ultracentrifugation at 167,000 g for 1 h to separate the membrane fraction. The membrane pellet was solubilized with 15 mM *n*-dodecyl-β-ᴅ-maltopyranoside (DDM, Anatrace) in a buffer containing 200 mM NaCl and 20 mM HEPES, pH 8.0 (buffer A), supplemented with 0.05 mg/ml soybean polar extract (asolectin, Avanti Polar Lipids) and 0.05% CHS (Avanti Polar Lipids) for 2 h at 4 °C. Nonsolubilized material was removed by ultracentrifugation (167,000×g for 25 min). The supernatant containing the solubilized protein was incubated with TALON resin pre-equilibrated with buffer A, 1 mM DDM, 0.05 mg/ml asolectin, and 0.05% CHS for 2 h at 4 °C. The beads were then washed with 10 column volumes of buffer A, 1 mM DDM, 0.05 mg/ml asolectin, 0.05% CHS, and 35 mM imidazole. The GlyR protein was eluted with buffer A, 1 mM DDM, 0.05 mg/ml asolectin, 0.05% CHS, and 250 mM imidazole. Eluted protein was concentrated and applied to a Superose 6 column (GE healthcare) equilibrated with buffer A and 1 mM DDM. Fractions containing the GlyR pentamers were collected and concentrated to 0.5 mg/ml using 50-kDa MWCO Millipore filters (Amicon).

The nanodisc reconstitution was carried out with the membrane scaffold protein (MSP1E3D1). MSP1E3D1 was expressed and purified with some modifications to the previously described protocols[19]. The MSP1E3D1 gene in pET 28a (a gift from Stephen Sligar: Addgene plasmid # 20066) was expressed in *E. coli* BL21(DE3) cells grown in Terrific broth medium supplemented with kanamycin (25 μg mL⁻¹) and 0.2% glucose. The culture was grown at 37 °C with shaking to an OD$_{600}$ of ~1.0, and induced by with 1 mM IPTG for 4 h at 37 °C. The cell pellet was lysed in a buffer containing 100 mM NaCl, 20 mM Tris base, pH 7.4 and supplemented with 1 mM PMSF and complete EDTA-free protease inhibitor cocktail tablet (Roche). The lysate was centrifuged at 30,000×g for for 30 min and the supernatant was bound to Ni-NTA, washed with four bed volumes of buffer (buffer B: 300 mM NaCl, 40 mM Tris, pH 8.0) and 1% Triton X-100, four bed volumes of buffer with 50 mM sodium cholate, four bed volumes of buffer B, four bed volumes of buffer B containing 20 mM imidazole, and eluted with buffer B containing 300 mM imidazole. The eluted MSP1E3D1 was passed through a desalting column equilibrated with 100 mM NaCl, 50 mM Tris, 0.5 mM EDTA, and pH 7.5, and the concentration was determined by absorbance at 280 nm (extinction coefficient = 29,910 M⁻¹ cm⁻¹). The purity was assessed by sodium dodecyl sulfate polyacrylamide gel electrophoresis and size-exclusion chromatography.

For reconstitution, the soybean polar lipid extract (Avanti Lipids) was dried in a stream of nitrogen and equilibrated in buffer A with protein, DDM, and MSP1E3D1 such that final molar ratio of Protein: MSP: lipid: DDM was 1:3:120:5. The mixture was incubated for 1 h with gentle rotation. After 1-h incubation, Bio-bead SM-2 (Bio-Rad Laboratories) were added and mixture was subjected to gentle rotation for 12 h at 4 °C. The reconstituted protein was applied to a Superose 6 column (GE healthcare) equilibrated with buffer A. Fractions containing protein nanodiscs were collected and concentrated to 0.1 mg/ml using 50-kDa MWCO Millipore filters (Amicon) for cryo-EM studies.

**Preparation of sample cryo-EM imaging and data acquisition.** GlyR nanodiscs samples from gel filtration (~0.1 mg/ml) was filtered and used for cryo-EM. For the Apo condition, the sample was used as such without any ligand (GlyR–Apo). For GlyR–Gly and GlyR–Gly/PTX, the samples were incubated for 1 h with 5 mM glycine and a combination of 5 mM glycine and 3 mM picrotoxin (PTX), respectively. For GlyR–Gly/IVM the samples were incubated for 1 h with either 1 mM glycine and 0.5 μM Ivermectin (IVM) or 0.1 mM glycine and 30 μM IVM. The sample was blotted

thrice with 3.5 μl sample each time onto glow-discharged Cu 300 mesh Quantifoil 1.2/1.3 grids (Quantifoil Micro Tools) and the grids were plunge frozen into liquid ethane using a Vitrobot (FEI). The grids were imaged using a 300 kV FEI Titan Krios microscope equipped with a Gatan K2-Summit direct electron detector camera (GlyR–Apo and GlyR–Gly/PTX datasets) or Gatan K3 direct electron detector camera for GlyR–Gly and GlyR–Gly–IVM datasets. The parameters for data acquisition for the different conditions are as follows. GlyR–Apo: Movies containing 40 frames were collected at 130,000× magnification (set on microscope) in super-resolution mode with a physical pixel size of 1.064 Å/pixel, dose per frame 1.30 e⁻/Å². Defocus values of the images ranged from −1.5 to −2.5 μm (input range setting for data collection) as per the automated imaging software Latitude (Gatan). GlyR–Gly/PTX: 40 frames/movie were collected with EPU (ThermoFisher Scientific) at 81,000× magnification (set on microscope) in counting mode with a physical pixel size of 1.06 Å/pixel, dose per frame 1.45 e⁻/Å². Defocus values ranged from −1.5 to −2.5. GlyR–Gly: 40 frames/movie were collected using Latitude at 81,000× magnification (set on microscope) with a physical pixel size of 1.08 Å/pixel, dose per frame 1.25 e⁻/Å². Defocus values ranged from −1.5 to −2.5 μm. GlyR–Gly–IVM: For the dataset with 1 mM glycine and 0.5 μM Ivermectin, movies containing 40 frames were collected at 81,000× magnification (set on microscope) in super-resolution mode with a physical pixel size of 1.08 Å/pixel, dose per frame 1.25 e⁻/Å². Defocus values of the images ranged from −1.5 to −2.5 μm (input range setting for data collection) as per the automated imaging software EPU. For the 0.1 mM glycine and 30 μM Ivermectin, movies containing 40 frames were collected with Latitude at 105,000× magnification (set on microscope) in super-resolution mode with a physical pixel size of 0.84 Å/pixel, dose per frame 1.25 e⁻/Å². Defocus values of the images ranged from −1.5 to −2.25 μm (input range setting for data collection).

**Image processing.** MotionCor2[48] (v 1.2.3) with a B-factor of 300 pixels[2] was used to correct beam-induced motion. Super-resolution images (for GlyR–Apo and GlyR–Gly datasets) were binned (2 × 2) in Fourier space, making a final pixel size of 1.064 and 1.08 Å, respectively. Initially, a small subset of micrographs were CTF-corrected with CTFFIND4[49] and processed in cisTEM[50] to generate 2D-templates for autopicking. All subsequent data processing was conducted in RELION 3.0[51]. The defocus values of the motion-corrected micrographs were estimated using Gctf software[52].

For GlyR–Apo dataset: 2, 042 micrographs of a total of 2, 389 were manually sorted. Totally, 444, 921 particles were auto-picked from 2, 042 micrographs with templates generated in cisTEM and were subjected to 2D classification to remove suboptimal particles. Totally, 61,000 good particles were selected and subjected to initial round of 3D auto-refinement using 30 Å low-pass filtered map of related human synaptic α1β3γ2 GABA$_A$R (EMD-4411) from which Megabody38 was manually erased in chimera[9]. Multiple rounds of 2D classification and 3D classification without image alignment were done to remove low resolution and broken particles. All the particles corresponding to bottom views of the receptor were removed from subsequent processing to prevent misalignment of the particles in 3D auto-refinement. The nanodisc belt was subtracted to improve the angular accuracy of alignment and remove sub-optimal particles. A set of 22, 643 particles resulted in a 3.7 Å map, which was used for training parameters for Bayesian polishing[51,53]. Optimized parameters (s_vel 1.029 --s_div 9585 --s_acc 2.805) were then used to polish these particles and the polished particles were then subjected to multiple rounds of 3D auto-refinement and 3D classification without image alignment. Multiple rounds of CTF refinement were performed before the final round of 3D auto-refinement. A final subset 19, 653 particles used for auto-refinement, with soft mask accounting for density of nanodisc, resulted in 3.39 Å map which was post-processed resulting in final resolution of 3.33 Å (Fourier shell coefficient (FSC) = 0.143 criterion). The B-factor estimation and map sharpening were performed in the post-processing step. Local resolutions were estimated using the RESMAP software[54].

For GlyR–Gly dataset, a total of 5344 movies were used. For autopicking in RELION 3.0, the final 2D class from the GlyR–Apo dataset was used as template and a total of 466,174 were auto-picked. Multiple rounds of 2D classification, 3D classification without image alignment was done for datasets. All the particles corresponding to the bottom views of receptor were removed from subsequent processing to prevent misalignment of the particles in 3D auto-refinement. A total of 15,000 particles were subjected to independent 3D auto refinement using GlyR–Apo map low-pass filtered to 20 Å as a template. The final ~10,000 particles were used for training parameters for Bayesian polishing. Multiple rounds of CTF refinement and classification were performed before the final round of 3D autorefinement. This resulted in a final map containing 8255 particles at 3.47 Å after post-processing. The B-factor estimation and map sharpening were performed in the post-processing step. Local resolutions were estimated using the RESMAP software[54].

For GlyR–Gly/PTX: a total of 2, 280 movies, collected on counting mode, were used from two imaging sessions. The grids for these imaging sessions were from the same batch, collected on the same microscope using identical parameters. For auto-picking in RELION3.0, the final 2D classes from the GlyR–Apo dataset was used as template and a total of 440,516 particles were picked. Multiple rounds of 2D classification and 3D classification without image alignment was used to get rid of low resolution particles. All the particles corresponding to the bottom views of receptor were removed from subsequent processing to prevent misalignment of the

particles in 3D auto-refinement. A total of 8, 288 particles from dataset 1 (1, 470 movies) and 11, 331 particles from dataset 2 (810 movies) were subjected to independent 3D autorefinement using GlyR–Apo map low pass filtered to 20 Å as a template. The two datasets were individually used for Bayesian polishing and the final polished datasets were merged in RELION 3.0.6. Few rounds of CTF refinement was performed before the final round of 3D autorefinement. The region corresponding to unstructured ICD was subtracted using RELION 3.0.6 and subtracted particles were further classified. The merged dataset with 12,503 particles was used for 3D autorefinement and further classification and resulted in a final map containing 10, 375 particles at 3.51 Å after post-processing. The B-factor estimation and map sharpening were performed in the post-processing step. Local resolutions were estimated using the RESMAP software[54].

For GlyR–Gly–IVM dataset with 1 mM glycine and 0.5 μM IVM, a total of 9700 movies were used. For autopicking in RELION 3.1, the final 2D class from the GlyR–Apo dataset was used as a template and a total of 860, 560 particles were auto-picked. A total of 45,000 particles were subjected to independent 3D auto refinement using GlyR–Apo map low pass filtered to 20 Å as a template. Multiple rounds of 2D classification, 3D classification without image alignment was done for datasets. All the particles corresponding to bottom views of the receptor were removed from subsequent processing to prevent misalignment of the particles in 3D autorefinement. The final ~35,000 particles were used for training parameters for Bayesian polishing. Few rounds of CTF refinement and 3D-classification were performed before the final round of 3D auto-refinement. This resulted in two distinct classes with 19,600 and 15,035 particles which resulted in 3.14 and 3.34 Å final maps for GlyR–Gly/IVM-1 and GlyR–Gly/IVM-2, respectively after post-processing. The B-factor estimation and map sharpening were performed in the post-processing step.

For GlyR–Gly–IVM dataset with 0.1 mM glycine and 30 μM Ivermectin, a total of 2,193 movies were used. For autopicking in RELION 3.1beta, the final 2D classes from the GlyR–Apo dataset was used as template and a total of 288,811 were auto-picked. A total of 45,000 particles were subjected to independent 3D auto refinement using GlyR–Apo map low pass filtered to 20 Å as a template. All the particles corresponding to the bottom views of receptor were removed from subsequent processing to prevent misalignment of the particles in 3D auto-refinement. The final ~30,000 particles were used for training parameters for Bayesian polishing. Few rounds of CTF refinement and 3D-classification were performed before the final round of 3D autorefinement. A final set of 27,516 particles was used for autorefinement which resulted in a 3.01 Å final map after post-processing. The B-factor estimation and map sharpening were performed in the post-processing step. Local resolutions were estimated using the RESMAP software[55].

**GlyR model building**. The 3D cryo-EM maps for the GlyR–Apo, GlyR–Gly, GlyR–Gly/PTX, GlyR–Gly/IVM datasets used for model-building contained density for the entire ECD, TMD, and a small region of the ICD. The final refined model of GlyR–Apo comprised of residues Pro31–Phe341 and Lys394–Gln444. The missing region (342–393) is of the unstructured ICD. The previously solved structure of GlyR–strychnine (PDB ID: 3JAD) was used as a starting model for the GlyR–Apo data set. The residues were renumbered as per the sequence available in UniProt database (O93430). For model building of GlyR–Gly, GlyR–Gly/PTX, GlyR–Gly–IVM data sets, the GlyR–Apo structure was used as a template. The M3 and M4 helices were truncated based on the quality of density in the respective maps for GlyR–Gly and GlyR–Gly–PTX. The cryo-EM map was first converted to the.mtz format using CCP4i software[56] with mapmask and sfall tools and then used for manual model building in COOT[57]. After initial model building, the three states were refined against their corresponding EM-derived maps using the phenix. real_space_refinement tool from the PHENIX software package[58], using rigid body, local grid, NCS, and gradient minimization. The individual models were then subjected to additional rounds of manual model fitting and refinement. The refinement statistics, the final model to map cross-correlation evaluated using phenix module mtriage[59], and the stereochemical properties of the models as evaluated by Molprobity[55] are detailed in the Supplementary Tables 3 and 4. Protein surface area and interfaces were analyzed using the PDBePISA server (http://www.ebi.ac.uk/pdbe/pisa/). Buried surface area refers to the solvent-accessible surface area of monomeric units that gets buried upon assembly formation, in this case a pentamer. PDBePISA server is used to calculate buried surface area (or water inaccessible regions – to a 1.4 Å diameter water probe) as a way to assess changes in interaction interfaces. The pore profile was calculated using the HOLE program[60]. The cavities were analyzed using Fpocket algorithm[61]. Figures were prepared using PyMOL v.2.0.4 (Schrödinger, LLC), CorelDraw v.20.1.0.708).

**MD simulations**. Molecular structures of the GlyR in various conformational states were separately embedded within phospholipid (POPC, 1-palmitoyl-2-oleoyl-sn-glycero-3-phosphocholine) bilayer membranes that were solvated on either side to form $13 \times 13 \times 17$ nm$^3$ simulation cells. Each protein-membrane system was assembled and equilibrated via a multiscale protocol[62]. Simulations were performed with GROMACS 2018[63], using the TIP4P/2005 water model[64] and the OPLS all-atom protein force field with united-atom lipids[65]. The integration time-step was 2 fs. Bonds were constrained through the LINCS algorithm[66]. A Verlet cut-off scheme was applied, and long-range electrostatic interactions were

calculated using the Particle Mesh Ewald method[67]. Temperature and pressure were maintained at 37 °C and 1 bar during simulations, using the velocity-rescaling thermostat[68] in combination with a semi-isotropic Parrinello and Rahman barostat[69], with coupling constants of $\tau_T = 0.1$ ps and $\tau_P = 1$ ps, respectively.

Pore water free energy profiles were computed for alternative conformations of the protein using the Channel Annotation Package[32], in each case based on three replicates of 30 ns equilibrium simulations at physiological salt (150 mM NaCl) concentration. To preserve the conformational state of each cryo-EM structure (while permitting rotameric flexibility in amino acid side chains) during simulations, harmonic restraints at a force constant of 1000 kJ mol$^{-1}$ nm$^{-2}$ were placed on protein backbone atoms. Simulation trajectories were analyzed at 500 ps intervals, with a bandwidth of 0.14 nm applied for water density estimation.

Chloride conduction was measured in five replicates of 200 ns simulations for each (backbone-restrained) receptor structure, at 500 mM NaCl concentration and in the presence of a +500 mV transmembrane potential difference, with positve potential on the cytoplasmic side. This was applied by imposing an external, uniform electric field in the membrane normal direction. Conductance values were calculated from the number of permeation events and averaged between replicates.

A separate set of equilibrium simulations (at 150 mM NaCl, in the absence of a transmembrane potential difference) were used to monitor the behavior of each protein conformation by sequential relaxation followed by removal of positional restraints. During an initial 10-ns simulation, harmonic restraints (again with force constant 1000 kJ mol$^{-1}$ nm$^{-2}$) were placed on all non-hydrogen atoms of the protein. Restraints on side chain atoms were then released, and the protein subjected to a further 20 ns of simulation. In the next 20 ns, only the α-carbon atom of each residue remained under the positional restraining force. The final state of the system was subsequently simulated for 200 ns, unrestrained.

**Reporting summary**. Further information on research design is available in the Nature Research Reporting Summary linked to this article.

## Data availability

Data supporting the findings of this paper are available from the corresponding authors upon reasonable request. A reporting summary for this article is available as a Supplementary Information file.

The atomic coordinates and cryo-EM maps for the GlyR–Apo, GlyR–Gly, and GlyR–Gly/PTX structures have been deposited in the Protein Data Bank and Electron Microscopy Data Bank with accession codes PDB-6UBS (EMD-20714), PDB-6UBT (EMD-20715), and PDB-6UD3 (EMD-20731), respectively. The atomic coordinates and cryo-EM maps for the GlyR–Gly-IVM-1, GlyR–Gly-IVM-2, and GlyR–Gly-IVM-3 structures have been deposited in the Protein Data Bank and Electron Microscopy Data Bank with accession codes PDB-6VM0 (EMD-21234), PDB-6VM2 (EMD-21236), and PDB-6VM3 (EMD-21237), respectively.

PDB 6UBS [https://doi.org/10.2210/pdb6UBS/pdb]
PDB 6UBT [https://doi.org/10.2210/pdb6UBT/pdb]
PDB 6UD3 [https://doi.org/10.2210/pdb6UD3/pdb]
PDB 6VM0 [https://doi.org/10.2210/pdb6VM0/pdb]
PDB 6VM2 [https://doi.org/10.2210/pdb6VM2/pdb]
PDB 6VM3 [https://doi.org/10.2210/pdb6VM3/pdb]

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

## Acknowledgements

We acknowledge the use of instruments at the National Cryo-Electron Microscopy Facility at the NCI and Stanford-SLAC Cryo-Electron Microscopy Facility. A special thanks to Prof. Wah Chiu for providing us imaging time and supervising the data collection (supported by National Institutes of Health grants: P41GM103832 and S10OD021600). We are grateful to the Cryo-Electron Microscopy Core at the CWRU School of Medicine and Dr. Kunpeng Li for the access to the sample preparation and Cryo-EM instrumentation. We thank Dr. Walter F. Boron for kindly providing us *Xenopus* oocytes and for unrestricted access of the oocyte rig. We are deeply appreciative of the support provided by Dr. Fraser Moss and Mr. Brian Zeise with the oocyte rig. We are very grateful to the members of the Chakrapani lab for critical reading and comments on the manuscript. This research was, in part, supported by the National Cancer Institute's National Cryo-EM Facility at the Frederick National Laboratory for Cancer Research. This work was supported by the National Institutes of Health grants R01GM108921, R01GM131216, R35GM134896, and Cryo-EM supplements: 3R01GM108921-03S1, R01GM108921-5S1, 3R01GM131216-1S1 to S.C. and the AHA postdoctoral Fellowship to A.K. (20POST35210394) and S.B. (17POST33671152).

## Author contributions

A.K. and S.C. conceived the project and designed the experimental procedures. A.K. purified the protein, and with assistance from S.B., optimized the cryo-EM sample preparation and performed grid screening. A.K carried out cryo-EM data analysis, model building, and refinement with inputs from S.B. M.L.M. collected the cryo-EM data for GlyR–Gly/PTX. A.K. and Y.G. performed two-electrode voltage-clamp recordings. S.R. performed the MD simulations, under the supervision of M.S. S.C. supervised the execution of the experiments, data analysis, and interpretation. A.K. and S.C. drafted the paper with contributions from S.B., S.R., and M.S. All authors reviewed the final paper.

## Competing interests

The authors declare no competing interests.
