## [Peer Review File · Nature Communications]

Peer Review File - Reviewers' comments first round:

Reviewer #1 (Remarks to the Author):

BACKGROUND FOR REVIEW

I have previously reviewed a former version of this manuscript by Kumar et al. submitted to another Nature journal. In my review of the present version of the manuscript, I have both reviewed the manuscripts in its present form and considered the changes introduced to the manuscript and the authors' rebuttal in response to my review of the former version of the manuscript. Comments or criticism of the previous version of the manuscript that I in this review have assessed to have been satisfactorily addressed by the authors are not mentioned in this review. I have performed this review and assessment of the manuscript with its submission to Nature Communications in mind.

REVIEW

The manuscript by Kumar et al. presents the determination of structures of the alpha1 glycine receptor in resting (GlyR-Apo), open (GlyR-Gly/PTX) and desensitized (GlyR-Gly) conformations using cryo-EM. The cryo-EM structures (resolutions 3.3-3.6Å) are of the full-length GlyR reconstituted in lipid nanodiscs, and the functional states of the structures are assigned and validated by molecular dynamics simulations. Based on the structures and analysis of them, the authors present the overall structure of the pentameric complex, the molecular compositions of specific regions/domains (the binding sites of Gly and PTX, the ion pore, the ICD, the ECD/TMD interface and the TMD, with particular focus on intermolecular interactions formed by M4 residues) and the global conformational changes in the receptor complex associated with activation and desensitization. The authors also present GlyR-Gly/ivermectin structures (non-conducting) and propose that these structures are intermediate to the open and desensitized GlyR structures. While numerous structures of GlyRs and other pLGICs have been published over the years, of particular importance for this work is that the authors through the use of full-length receptor and lipid nanodiscs have obtained an "open" conformation of the receptor (GlyR-Gly/PTX) that is more believable than the "super-open" conformation previously published.

The data presented in the manuscript is original and novel, and the analysis of the available structures of resting, open and desensitized conformations does facilitate an in-depth insight into the conformational transitions of the GlyR and the molecular mechanisms underlying activation and desensitization. However, much of the information gained from this study is more or less obtainable from previous GlyR structures or from previous studies delineating allosteric transitions between different receptor conformations for other pLGICs. This is not a criticism of the performed work in this study, and the obtained structures certainly will be of interest, but as a reader one is perhaps too often left with the impression that this is verification of previous findings rather than true new insights into GlyRs and pLGICs.

As for the techniques and methodologies applied to address the research questions in this work, the structure determinations/analysis and molecular dynamics simulations performed are advanced and appropriate for this kind of work. The generated data is of high quality, and data analysis and statistical analysis appear to have been done consistently and correctly. However, with the exception of a few simple TEVC recordings verifying the functional properties of glycine and picrotoxin as an agonist and a channel blocker, respectively, at GlyR expressed in oocytes, no electrophysiological studies of observations made from the structures have been performed. As outlined specifically below, this lack of validation/verification of functional implications of observed intermolecular interactions represents a flaw in this nice study.

The research questions, the rationale for the experimental design applied, the generated data and the interpretation of data are largely presented in a logical and precise manner in the manuscript, where appropriate credit is given to previous work in the field.

CONCLUSION

This is a very nice study that provides high-quality structures of the GlyR in its three major conformations as well as additional proposed intermediate conformations using advanced cryo-EM and molecular dynamics techniques.

When it comes to my three major concerns about the former version of this manuscript submitted to another Nature journal, two of these have been satisfactorily addressed by the authors in the present manuscript submitted to Nature Communications:

1) While it is still this reviewers' opinion that considerable parts of the information provided by this work are extractable from previous structural biology studies of GlyR and other pLGICs, the work does provide novel and interesting findings about an important neurotransmitter receptor and receptor class. Thus, as for this criteria I think it warrants publication in Nature Communications.

2) The authors make a much stronger case for the GlyR-Gly/PTX structure being a valid surrogate structure for the open conformation of GlyR than in the former version of the manuscript, although I think it is very important that they include their arguments for this explicitly in the manuscript.

3) My concern about the lack of functional studies validating/verifying observations made in the structures has not been addressed in the present version of the manuscript. As argued in the specific comment below, I do believe that the arguments by the authors against this concern are particularly valid. I think that this lack of validation/verification of these structural details is highly unfortunate since it renders these potentially interesting findings much more vague and non-conclusive than they could – and should – be. Moreover, on this foundation the authors risk that their suggestion of these molecular interactions being important for GlyR function eventually could be contradicted by subsequent studies.

All in all, my overall assessment of the work is that this is a study that will be of considerable interest to the GlyR and pLGIC fields and to the neuroscience and structural biology fields in general.

SPECIFIC COMMENTS

"Abstract"

- line 6: maybe "GlyR protein complex" instead of "GlyR protein"?
- line 12: maybe "single-channel conductance properties"?

"Introduction"

- p.2, line 1 and 5-6: The abbreviations of receptor should be uniform. The authors have three different abbreviation styles: "glycine receptors (GlyRs)", "GABAA receptors (GABAA)", "nicotinic acetylcholine receptors (nAChR)" and "5HT3 receptors (5HT3R)". Be consistent about including an "R" and a plural "s" for "receptors" in the abbreviations.
- P.2, line 18: it should be "channel" instead of "channels".

"Results and Discussion"

- P.5, first paragraph, P.10-11, p.15-17: One of my principle concerns regarding the manuscript in its former version was the use of a Gly- and PTX-bound structure to reflect the open conformation of GlyR. While I understood the rationale for this approach and the need to obtain the channel blocker-bound structure to access the putative open conformation, the authors were asked to argue for the validity of the structure for this purpose. In their rebuttal, the authors convincingly argue for this point, including outlining the essentially identical 9', 6' and 2' regions in the GlyR-Gly and GlyR-Gly/PTX structure, and they have included new GlyR-ivermectin structures that are proposed to represent intermediate states between the open and the desensitized GlyR. PTX is observed not to induce changes in its binding region and the authors conclude that it acts allosterically on the intracellular P-2' gate and prevents it from closing. In this reviewers' opinion, this still raises the question whether this intracellular gate in its open conformation in the GlyR-Gly/PTX structure has

to be identical to that of the "true" open GlyR conformation. In the "true" open GlyR, the channel is open because the collapse of the channel in the course of the desensitization process has not happened yet, in GlyR-Gly/PTX structure the channel is open because the bound ligand prevents this collapse. While I fully agree with the authors that structural arrangement of also this part in the GlyR-Gly/PTX structure very likely is a good representation of it in the "true" open GlyR, I do not think one can be absolutely sure.

In conclusion, I find the authors' explanations/arguments and the added experimentation to be very solid, and thus the authors have been very responsive to this comment. The only thing I would ask for here is that the good arguments provided for the Gly/PTX-GlyR structure as the "open" conformation in the rebuttal also are presented in their discussion of the structure in the manuscript text. I think it would be important to convey the use of this structure as a "surrogate" for the open conformation and to substantiate the validity of this use as such to the reader, and when the authors have so strong arguments as they have, it only strengthens their case for this claim.

- p.5, lines 6-12: It would be good with a reference to Supplemental Figure 1 in this segment with the effects of glycine and picrotoxin are outlines. It comes later in the paragraph but then it is a reference for the strategy for the structures.
- General comment: Picrotoxin is an equimolar mixture of picrotonin and picrotoxinin. The authors appear to have used the latter compound in for example Fig. 2D-E, which is fine since it is the more potent GlyR inhibitor. It is also fine to refer to the structures at picrotoxin, but somewhere in the manuscript there should be a mentioning of this, since there strictly speaking is no such thing as a picrotoxin molecule.
- p.6, 3rd line from bottom: It should be "pLGICs"
- p. 12, top segment & p. 14, bottom segment: The Å values should be given without the space between the first two and the last three digits.
- p.12 bottom half – p. 13, lines 1-6 & Fig. 4: Another principle concerns regarding the manuscript in its former version was the lack of investigations of the functional importance of any of the many interactions between residues identified in specific regions of the GlyR structures by electrophysiological recordings (interactions shown in Fig. 4A-B in this manuscript, and in Fig. 5 in the former version). The authors have not included any data on this in the present version of the manuscript.

The authors argue in their rebuttal that they have identified an extensive network of interactions at interdomain interfaces that are altered between different conformational states, and that they in the present version of the manuscript have highlighted residues previously investigated in functional studies of the hyperekplexia-causing mutations and included references to these studies. I certainly acknowledge that, but these additions are all given with regard to the ECD-ICD cross-interactions shown in Fig. 3, and not for those interactions that my comment was regarding, those formed residues in post-M4 to beta8/9- and pre-M1-residues (presented in Fig. 4A in the present manuscript) and those formed between M4-preM4 residues and lipid molecules (presented in Fig. 4B in the present manuscript). The authors do not themselves validate or present previous studies validating the functional implications of these interactions. In the case of the latter interactions, they state that these investigations are ongoing but argue that the results from these can not be included in the manuscript due to its length.

In my opinion, this segment and the findings conveyed in Figs. 4A-B constitutes the weakest part of this nice study, and I would argue that electrophysiological verification of selected observations made in ones structures are to be expected. In their rebuttal the authors claim that detailed functional analysis is needed to investigate these functional implications, and while that may be true the authors could come a long way by studying the functional properties of mutants comprising mutation of the 6 or more residues (such as Asp218, Arg220, Glu 244, Arg437, Gln444, Arg406) identified to form these interactions. As perhaps best evidenced by the sentence immediately before this segment (p.12, lines 8-11), where the authors reference previous studies

to substantiate that the ECD-ICD cross-interaction have substantial functional implications, such as electrophysiological validation/exploration of the actual functional implications of these identified interactions would bring a whole other level of impact to these findings. The lack of such investigation makes the present findings purely observational and in effect renders the author unable to do anything beyond hand-waving when it comes the functional importance of these interactions.

- P.13, bottom third – P. 15, line 4, P. 17, Fig. 5 and Supp Fig 8-10: In this version of their manuscript the authors have included structures of ivermectin/Gly-bound GlyR structures, which in MD experiments are found to be non-conducting. In addition to the comparison of the basic features of these structures to previously published IVM-GlyR structures (the authors need to reference these structures and studies in this section) and to the Gly-GlyR and Gly/PTX-GlyR structures, the authors interestingly find a difference in the Pro-2' side-chains in these structures compared to Gly-GlyR and Gly/PTX-GlyR and propose that these structures could represent intermediate states to the desensitized and "open" conformations. These added structures and the analysis of them add to the impact of the study – and, specifically, also helps building the case for the validity of the Gly/PTX-GlyR structure for the "open" GlyR conformation.

Anders A. Jensen

Reviewer #2 (Remarks to the Author):

Kumar et al. present new structures of GlyR in lipid nanodiscs obtained in the presence of different ligands. Comparing the structures, the authors observe conformational changes in the protein and interpret them as representing functional gating dynamics, also complementary analyzed using MD simulations. While the study is important and interesting, there are some issues for authors to consider.

Starting from the title, the manuscript intends to oversale the significance of the protein preparation in nanodiscs. Nanodiscs are not membranes and there is no proof that they are any better for capturing membrane proteins in conformations more physiological than using detergents, amphipols or SMAs. Referencing one paper, even if it is published in Nature (Lavery et al., 2019), does not make the opinion about lipid nanodiscs as a mimic of natural lipid environment "generally accepted". The title therefore should be changed to refer to "glycine receptor in lipid nanodiscs". Similarly, "in a physiological environment" should be removed from the last sentence of the Abstract.

The quality of models should be significantly improved. Not only the side chains do not often fit the density (very apparent for the bulky side chains of Arg, Lys and aromatic residues) but even the backbone is often traced incorrectly. For many regions of the models, maps are extremely weak or missing, especially for the extracellular or intracellular ends of the peripheral transmembrane helices. Precision of models in these regions must be critical for estimating intra-subunit and inter-subunit cavities illustrated in Fig. 4C. Similarly, many of the modelled sugars are not supported by the densities or show extremely poor fit. It is understandable that there were probably more complete structures available from before but for each of these new presented structures, the models need to be trimmed so that they only contain what is visible in density.

There are lipid molecules fitted into non-protein densities surrounding the TM region. The quality of these densities is way too poor to unambiguously fit any lipid or detergent molecules and surely not good enough to specify the type of lipid/detergent. The corresponding models of lipids have to be removed from the corresponding PDBs. If authors want to illustrate the non-protein densities surrounding the TM region, they can show them in cryo-EM maps (alongside the protein density at the same contour level) and call "lipid-like" densities.

Glycine molecule in the ligand binding pocket fits a shapeless blob of density. The fit of glycine molecule is therefore based on the chemical environment of the binding site. While some of glycine molecules are fitted with the carboxyl group of glycine facing R89 as anyone would expect, others

are flipped 180 degrees with the carboxyl group facing away. Is there experimental evidence for such drastically different placement?

What is considered the density for PTX is at the level of noise and cannot be seriously considered a reason for placing the molecule of PTX at the corresponding location. The authors might think of repeating the corresponding experiment, may be at a high PTX concentration, in order to convince the reader that the corresponding binding site is real. After PTX is modeled correctly, it will be worth making side-by-side comparisons with GluCl, compared to which the current binding site is much higher, and GABAA, where the location is also a bit different.

The Aricescu GABAA receptor pore in the presence of PTX adopts a conformation essentially identical to the resting state, which is quite different from that observed here with the GlyR. Many studies have been performed with PTX on glycine receptors; it would be helpful to cite some of them. In Suppl. Fig. 1, if PTX blocks desensitization as stated in the text, why is the channel more desensitized after PTX application than before?

The issue of the previously reported widely open state has become quite contentious with many groups working to resolve this question. There are conflicts even between the computational groups. It is important to not overstate the strength of the data in either direction to help get to the right answer eventually. In the Introduction, the authors suggest that truncation of the ICD may explain why the Gouaux open state from 2015 was so wide and they cite two papers they suggest support this idea: "The ICD, besides its role in trafficking, has also been implicated to alter single-channel conductance and gating behavior (refs. 3, 25)." However, these references have nothing to do with the ICD. Is there actually evidence that the truncation causes any sort of functional behavior change in anionic pLGICs?

In single particle processing for the open state, did the authors find and exclude 3D classes with widely open pores?

The more than 4-fold difference in conductance predicted by MD simulations (20 pS) versus the experiment (80-88 pS, ref. 60) is explained by missing residues in the ECD (page 16). However, the most significant effect of mutating the basic cluster of residues in the ECD was only a change from 92 to 60 pS (ref. 61). Please expand a bit the discussion on computational conductance measurements vs. those from single-channel experiments. How good now is computational electrophysiology at predicting/nailing down real physiological values? Is it possible for MD to make any realistic predictions, when positional restraints are placed on the protein backbone during the simulations (page 15), while the transmembrane potential is +500 mV and the sodium concentration is 500 mM (page 16), which are so much different from physiological values?

Minor:

Page 3, third line from the bottom: "preformed" should be removed.

Page 5, first sentence. The sentence should be rewritten: authors would like to believe that the conditions of their experiment favor physiological conditions but they cannot be so sure of this.

Page 10, last line. There is no panel D in Figure 3.

Why are the initial particle numbers approximate in Suppl. Table 1?

Are the map sharpening B factors correct? They seem very small (in absolute value).

Reviewer #3 (Remarks to the Author):

In this work, structural basis for gating, ion selectivity, and single-channel conductance of GlyR in membrane was investigated based on the cryo-EM structures and molecular dynamics simulations. I particularly evaluate molecular dynamics simulation parts of this work.

The molecular dynamics simulations were performed with standard computational technique. The comparison between the simulations and the cryo-EM structures is scientifically meaningful. However, there are a number of uncertain points for this reviewer, which should be clarified in a revised manuscript.

Main Comments:

1) In Page 12, the authors suggested that the interactions between subunits were changed, and the overall buried surface area was decreased in the GlyR-Gly/PTX and GlyR-Gly. However, relationship between interaction change and surface area change is not so clearly described. Why did the authors quantify the overall surface area change? And also, what is the "buried surface" area?

2) In Page 17, the authors described that the pore features of the three conformations were stably maintained in the MD simulations without positional restraints. Is the pore still dewetted at Leu9' in Gly-Apo or wetted in GlyR-Gly/PTX and GlyR-Gly, as shown in Figure 6C?

Minor comments:

1) Typo: In Pages 10 and 11, Fig3D and Fig 3E should be Fig 2D and Fig2E. There is no explanation about Fig2F in the main text.

2) Panels in all figures are very busy. For example, there is a clash between Panels A and B in Figure 1. Position of all panels should be rearranged well.

3) It is difficult to recognize some labels in the figures (e.g., Figure 2C, Panel A (right) in Figure 3, and Figure 4)

We thank all the three Reviewers for their constructive comments and criticisms. We have addressed these to the best of our ability, and we believe that the manuscript has significantly improved in response to the feedback. Substantial changes included in this revision:

- 1. Updated structural models with improved fits to the map (map resolution improved in some cases upon reprocessing the data). The new PDBs and maps have been deposited in RCSB and they are included here to be shared with the Reviewers.*
- 2. We have also imaged under higher concentration of PTX (5 mM) as suggested by Reviewer 2. The data is included for Reviewer evaluation. Since the structure is essentially the same as that of 3 mM PTX complex described in the paper, we have not included this information in the main text.*
- 3. We have carried out the MD simulations for the updated models.*
- 4. Additional discussion and other requested minor changes have been made.*

Reviewers' comments:

Reviewer #1 (Remarks to the Author):

BACKGROUND FOR REVIEW

I have previously reviewed a former version of this manuscript by Kumar et al. submitted to another Nature journal. In my review of the present version of the manuscript, I have both reviewed the manuscripts in its present form and considered the changes introduced to the manuscript and the authors' rebuttal in response to my review of the former version of the manuscript. Comments or criticism of the previous version of the manuscript that I in this review have assessed to have been satisfactorily addressed by the authors are not mentioned in this review. I have performed this review and assessment of the manuscript with its submission to Nature Communications in mind.

REVIEW

The manuscript by Kumar et al. presents the determination of structures of the alpha1 glycine receptor in resting (GlyR-Apo), open (GlyR-Gly/PTX) and desensitized (GlyR-Gly) conformations using cryo-EM. The cryo-EM structures (resolutions 3.3-3.6Å) are of the full-length GlyR reconstituted in lipid nanodiscs, and the functional states of the structures are assigned and validated by molecular dynamics simulations. Based on the structures and analysis of them, the authors present the overall structure of the pentameric complex, the molecular compositions of specific regions/domains (the binding sites of Gly and PTX, the ion pore, the ICD, the ECD/TMD interface and the TMD, with particular focus on intermolecular interactions formed by M4 residues) and the global conformational changes in the receptor complex associated with activation and desensitization. The authors also present GlyR-Gly/ivermectin structures (non-conducting) and propose that these structures are intermediate to the open and desensitized GlyR structures. While numerous structures of GlyRs and other pLGICs have been published over the years, of particular importance for this work is that the authors through the use of full-length receptor and lipid nanodiscs have obtained an "open" conformation

of the receptor (GlyR-Gly/PTX) that is more believable than the “super-open” conformation previously published.

The data presented in the manuscript is original and novel, and the analysis of the available structures of resting, open and desensitized conformations does facilitate an in-depth insight into the conformational transitions of the GlyR and the molecular mechanisms underlying activation and desensitization. However, much of the information gained from this study is more or less obtainable from previous GlyR structures or from previous studies delineating allosteric transitions between different receptor conformations for other pLGICs. This is not a criticism of the performed work in this study, and the obtained structures certainly will be of interest, but as a reader one is perhaps too often left with the impression that this is verification of previous findings rather than true new insights into GlyRs and pLGICs.

As for the techniques and methodologies applied to address the research questions in this work, the structure determinations/analysis and molecular dynamics simulations performed are advanced and appropriate for this kind of work. The generated data is of high quality, and data analysis and statistical analysis appear to have been done consistently and correctly. However, with the exception of a few simple TEVC recordings verifying the functional properties of glycine and picrotoxin as an agonist and a channel blocker, respectively, at GlyR expressed in oocytes, no electrophysiological studies of observations made from the structures have been performed. As outlined specifically below, this lack of validation/verification of functional implications of observed intermolecular interactions represents a flaw in this nice study.

The research questions, the rationale for the experimental design applied, the generated data and the interpretation of data are largely presented in a logical and precise manner in the manuscript, where appropriate credit is given to previous work in the field.

CONCLUSION

This is a very nice study that provides high-quality structures of the GlyR in its three major conformations as well as additional proposed intermediate conformations using advanced cryo-EM and molecular dynamics techniques.

When it comes to my three major concerns about the former version of this manuscript submitted to another Nature journal, two of these have been satisfactorily addressed by the authors in the present manuscript submitted to Nature Communications:

- 1) While it is still this reviewer's opinion that considerable parts of the information provided by this work are extractable from previous structural biology studies of GlyR and other pLGICs, the work does provide novel and interesting findings about an important neurotransmitter receptor and receptor class. Thus, as for this criteria I think it warrants publication in Nature Communications.
- 2) The authors make a much stronger case for the GlyR-Gly/PTX structure being a valid surrogate structure for the open conformation of GlyR than in the former version of the manuscript, although I think it is very important that they include their arguments for this explicitly in the manuscript.

3) My concern about the lack of functional studies validating/verifying observations made in the structures has not been addressed in the present version of the manuscript. As argued in the specific comment below, I do believe that the arguments by the authors against this concern are particularly valid. I think that this lack of validation/verification of these structural details is highly unfortunate since it renders these potentially interesting findings much more vague and non-conclusive than they could – and should - be. Moreover, on this foundation the authors risk that their suggestion of these molecular interactions being important for GlyR function eventually could be contradicted by subsequent studies.

All in all, my overall assessment of the work is that this is a study that will be of considerable interest to the GlyR and pLGIC fields and to the neuroscience and structural biology fields in general.

We are grateful to the Reviewer for taking the time to provide us a thorough review of our manuscript. Through several new experiments, included in the previous revision and in this one, and additional discussion, we think our manuscript has been significantly strengthened. Please see below for details.

SPECIFIC COMMENTS

“Abstract”

- line 6: maybe “GlyR protein complex” instead of “GlyR protein”?

Done

- line 12: maybe “single-channel conductance properties”?

Done

“Introduction”

- p.2, line 1 and 5-6: The abbreviations of receptor should be uniform. The authors have three different abbreviation styles: “glycine receptors (GlyRs)”, “GABAA receptors (GABAA)”, “nicotinic acetylcholine receptors (nAChR)” and “5HT3 receptors (5HT3R)”. Be consistent about including an “R” and a plural “s” for “receptors” in the abbreviations.

Done

- P.2, line 18: it should be “channel” instead of “channels”.

Done

“Results and Discussion”

• P.5, first paragraph, P.10-11, p.15-17: One of my principle concerns regarding the manuscript in its former version was the use of a Gly- and PTX-bound structure to reflect the open conformation of GlyR. While I understood the rationale for this approach and the need to obtain the channel blocker-bound structure to access the putative open conformation, the authors were asked to argue for the validity of the structure for this purpose. In their rebuttal, the authors convincingly argue for this point, including outlining the essentially identical 9', 6' and 2' regions in the GlyR-Gly and GlyR-Gly/PTX structure, and they have included new GlyR-ivermectin structures that are proposed to represent intermediate states between the open and the desensitized GlyR. PTX is observed not to induce changes in its binding region and the authors conclude that it acts allosterically on the intracellular P-2' gate and prevents it from closing. In this reviewer's opinion, this still raises the question whether this intracellular gate in its open conformation in the GlyR-Gly/PTX structure has to be identical to that of the "true" open GlyR conformation. In the "true" open GlyR, the channel is open because the collapse of the channel in the course of the desensitization process has not happened yet, in GlyR-Gly/PTX structure the channel is open because the bound ligand prevents this collapse. While I fully agree with the authors that structural arrangement of also this part in the GlyR-Gly/PTX structure very likely is a good representation of it in the "true" open GlyR, I do not think one can be absolutely sure.

In conclusion, I find the authors' explanations/arguments and the added experimentation to be very solid, and thus the authors have been very responsive to this comment. The only thing I would ask for here is that the good arguments provided for the Gly/PTX-GlyR structure as the "open" conformation in the rebuttal also are presented in their discussion of the structure in the manuscript text. I think it would be important to convey the use of this structure as a "surrogate" for the open conformation and to substantiate the validity of this use as such to the reader, and when the authors have so strong arguments as they have, it only strengthens their case for this claim.

Per suggestion, we have added this discussion to the main text in the paper (Pages 5 and 17) and is accompanied by Supplemental Figure 8 shows the comparison of the GlyR-Gly and GlyR-Gly/PTX pores.

• p.5, lines 6-12: It would be good with a reference to Supplemental Figure 1 in this segment with the effects of glycine and picrotoxin are outlined. It comes later in the paragraph but then it is a reference for the strategy for the structures.

Done

• General comment: Picrotoxin is an equimolar mixture of picrotonin and picrotoxinin. The authors appear to have used the latter compound in for example Fig. 2D-E, which is fine since it is the more potent GlyR inhibitor. It is also fine to refer to the structures at picrotoxin, but somewhere in the manuscript there should be a mentioning of this, since strictly speaking there is no such thing as a picrotoxin molecule.

We have now included this in page 5.

- p.6, 3rd line from bottom: It should be “pLGICs”

Done

- p. 12, top segment & p. 14, bottom segment: The Å² values should be given without the space between the first two and the last three digits.

Done

• p.12 bottom half – p. 13, lines 1-6 & Fig. 4: Another principle concerns regarding the manuscript in its former version was the lack of investigations of the functional importance of any of the many interactions between residues identified in specific regions of the GlyR structures by electrophysiological recordings (interactions shown in Fig. 4A-B in this manuscript, and in Fig. 5 in the former version). The authors have not included any data on this in the present version of the manuscript. The authors argue in their rebuttal that they have identified an extensive network of interactions at interdomain interfaces that are altered between different conformational states, and that they in the present version of the manuscript have highlighted residues previously investigated in functional studies of the hyperekplexia-causing mutations and included references to these studies. I certainly acknowledge that, but these additions are all given with regard to the ECD-ICD cross-interactions shown in Fig. 3, and not for those interactions that my comment was regarding, those formed residues in post-M4 to beta8/9- and pre-M1-residues (presented in Fig. 4A in the present manuscript) and those formed between M4-preM4 residues and lipid molecules (presented in Fig. 4B in the present manuscript). The authors do not themselves validate or present previous studies validating the functional implications of these interactions. In the case of the latter interactions, they state that these investigations are ongoing but argue that the results from these can not be included in the manuscript due to its length.

In my opinion, this segment and the findings conveyed in Figs. 4A-B constitutes the weakest part of this nice study, and I would argue that electrophysiological verification of selected observations made in ones structures are to be expected. In their rebuttal the authors claim that detailed functional analysis is needed to investigate these functional implications, and while that may be true the authors could come a long way by studying the functional properties of mutants comprising mutation of the 6 or more residues (such as Asp218, Arg220, Glu 244, Arg437, Gln444, Arg406) identified to form these interactions. As perhaps best evidenced by the sentence immediately before this segment (p.12, lines 8-11), where the authors reference previous studies to substantiate that the ECD-ICD cross-interaction have substantial functional implications, such as electrophysiological validation/exploration of the actual functional implications of these identified interactions would bring a whole other level of impact to these findings. The lack of such investigation makes the present findings purely observational and in effect renders the author unable to do anything beyond hand-waving when it comes the functional importance of these interactions.

We agree with the Reviewers concerns regarding the need for functional validation of the potential interactions observed in the structures. While we have highlighted numerous residues within the interaction network which when perturbed by a mutation leads to disease phenotype,

we do acknowledge that further characterization are needed to better mechanistic interpretation. In response to Reviewer 2's concern about the map density at the tips of M4, we have truncated these regions in the current model and removed the discussion of the interaction between M2-lipid and the postM4-ICD. We are hopeful that we will resolve these regions in the near future and investigate these questions further.

• P.13, bottom third – P. 15, line 4, P. 17, Fig. 5 and Supp Fig 8-10: In this version of their manuscript the authors have included structures of ivermectin/Gly-bound GlyR structures, which in MD experiments are found to be non-conducting. In addition to the comparison of the basic features of these structures to previously published IVM-GlyR structures (the authors need to reference these structures and studies in this section) and to the Gly-GlyR and Gly/PTX-GlyR structures, the authors interestingly find a difference in the Pro-2' side-chains in these structures compared to Gly-GlyR and Gly/PTX-GlyR and propose that these structures could represent intermediate states to the desensitized and “open” conformations. These added structures and the analysis of them add to the impact of the study – and, specifically, also helps building the case for the validity of the Gly/PTX-GlyR structure for the “open” GlyR conformation.

Thank you for noting this point. We agree, the three new IVM-bound conformations (with resolutions 3.0-3.3 Å) included in the revision provide additional information on the conformational transitions that link the open and desensitized conformations.

Reviewer #2 (Remarks to the Author):

Kumar et al. present new structures of GlyR in lipid nanodiscs obtained in the presence of different ligands. Comparing the structures, the authors observe conformational changes in the protein and interpret them as representing functional gating dynamics, also complementary analyzed using MD simulations. While the study is important and interesting, there are some issues for authors to consider.

Starting from the title, the manuscript intends to oversale the significance of the protein preparation in nanodiscs. Nanodiscs are not membranes and there is no proof that they are any better for capturing membrane proteins in conformations more physiological than using detergents, amphipols or SMAs. Referencing one paper, even if it is published in Nature (Lavery et al., 2019), does not make the opinion about lipid nanodiscs as a mimic of natural lipid environment “generally accepted”. The title therefore should be changed to refer to “glycine receptor in lipid nanodiscs”. Similarly, “in a physiological environment” should be removed from the last sentence of the Abstract.

We have made the change to the title and abstract as suggested. The sentence in the text referencing the use of nanodiscs is reworded “Furthermore, lipids are essential components of membrane protein stability, and lipid nanodisc derived from membrane scaffolding protein (MSP) is one such tool used to mimic the lipid bilayer environment¹⁷.”

The quality of models should be significantly improved. Not only the side chains do not often fit the density (very apparent for the bulky side chains of Arg, Lys and aromatic residues) but even the backbone is often traced incorrectly. For many regions of the models, maps are extremely weak or missing, especially for the extracellular or intracellular ends of the peripheral transmembrane helices. Precision of models in these regions must be critical for estimating intra-subunit and inter-subunit cavities illustrated in Fig. 4C. Similarly, many of the modelled sugars are not supported by the densities or show extremely poor fit. It is understandable that there were probably more complete structures available from before but for each of these new presented structures, the models need to be trimmed so that they only contain what is visible in density.

We have made the changes to the models, including trimming of the peripheral ends of M4, fixing poor fit issues, rebuilt shorter versions of lipids and glycans. Reprocessing the data has made small improvements in resolution and map quality. Please see the updated models and maps.

There are lipid molecules fitted into non-protein densities surrounding the TM region. The quality of these densities is way too poor to unambiguously fit any lipid or detergent molecules and surely not good enough to specify the type of lipid/detergent. The corresponding models of lipids have to be removed from the corresponding PDBs. If authors want to illustrate the non-protein densities surrounding the TM region, they can show them in cryo-EM maps (alongside the protein density at the same contour level) and call “lipid-like” densities.

As suggested, we call them lipid-like densities. The headgroups for the lipids are removed from the model. The protein and lipid densities contoured at the same level is shown in a new supplemental figure 6.

Glycine molecule in the ligand binding pocket fits a shapeless blob of density. The fit of glycine molecule is therefore based on the chemical environment of the binding site. While some of glycine molecules are fitted with the carboxyl group of glycine facing R89 as anyone would expect, others are flipped 180 degrees with the carboxyl group facing away. Is there experimental evidence for such drastically different placement?

Thank you for pointing it out. One of the IVM structures had an inverted Gly orientation. We have fixed this.

What is considered the density for PTX is at the level of noise and cannot be seriously considered a reason for placing the molecule of PTX at the corresponding location. The authors might think of repeating the corresponding experiment, may be at a high PTX concentration, in order to convince the reader that the corresponding binding site is real. After PTX is modeled correctly, it will be worth making side-by-side comparisons with GluCl, compared to which the current binding site is much higher, and GABAA, where the location is also a bit different.

In response to this criticism, we have now included the following:

- (i) Panel 2D includes the density for M2 and PTX contoured at the same level.*
- (ii) Supplemental Figure 9 shows a side-by-side comparison of map-model for PTX complexes with GlyR, GABA, and GluCl. (Additionally, we note positional differences*

in PTX in GlyR-Gly/PTX and GABA_AR ($\alpha 1\beta 3\gamma 2$) structures compared with GluCl (Supplemental Figure 9) where PTX interacts with Thr2' instead. We think this variation may arise from the nature of the sidechain at the 2' position (Gly in homomeric $\alpha 1$ GlyR, Val/Ala/Ser in $\alpha 1\beta 3\gamma 2$ GABA_AR, or a Thr in GluCl) or from slightly different mechanisms of PTX action as is also exemplified by use-dependent nature of PTX block in GABA_AR and GluCl but not in the case of GlyR).

- (iii) *Per suggestion we have imaged GlyR at a higher PTX concentration (3 mM Gly/5 mM PTX) and refined the 3 D reconstruction to 3.70 Å. Since we didn't see any notable differences in the map or model, we are including this information here in the response to reviewer comments. (Reviewer Figure 1 and 2).*

The Aricescu GABA_A receptor pore in the presence of PTX adopts a conformation essentially identical to the resting state, which is quite different from that observed here with the GlyR. Many studies have been performed with PTX on glycine receptors; it would be helpful to cite some of them. In Suppl. Fig. 1, if PTX blocks desensitization as stated in the text, why is the channel more desensitized after PTX application than before?

Per suggestion, additional references are included for the effect of PTX on GlyR. Previous studies have shown extensive characterization of the effect of PTX on inhibiting GlyR desensitization (Gielen et al). The representative traces are shown on same time axis with arrow highlighting the amount of current before and after PTX application.

The issue of the previously reported widely open state has become quite contentious with many groups working to resolve this question. There are conflicts even between the computational groups. It is important to not overstate the strength of the data in either direction to help get to the right answer eventually. In the Introduction, the authors suggest that truncation of the ICD may explain why the Gouaux open state from 2015 was so wide and they cite two papers they suggest support this idea: "The ICD, besides its role in trafficking, has also been implicated to alter single-channel conductance and gating behavior (refs. 3, 25)." However, these references have nothing to do with the ICD. Is there actually evidence that the truncation causes any sort of functional behavior change in anionic pLGICs?

We have elaborated this section with appropriated references. A recent report from the Sivilotti group that shows that ICD truncation in GlyR leads to an increase in maximum single-channel open probability and a substantial increase in agonist efficacy, particularly for partial agonists which elicit responses similar to full agonists (Ivica, et al , 2020).

In single particle processing for the open state, did the authors find and exclude 3D classes with widely open pores?

No, we did not find classes with wide-open pores.

The more than 4-fold difference in conductance predicted by MD simulations (20 pS) versus the experiment (80-88 pS, ref. 60) is explained by missing residues in the ECD (page 16). However,

the most significant effect of mutating the basic cluster of residues in the ECD was only a change from 92 to 60 pS (ref. 61). Please expand a bit the discussion on computational conductance measurements vs. those from single-channel experiments. How good now is computational electrophysiology at predicting/nailing down real physiological values? Is it possible for MD to make any realistic predictions, when positional restraints are placed on the protein backbone during the simulations (page 15), while the transmembrane potential is +500 mV and the sodium concentration is 500 mM (page 16), which are so much different from physiological values?

The Reviewer makes a fair point. We have re-run the computational electrophysiology simulations to estimate the conductance, using the slightly improved (in terms of resolution and model building) GlyR+PTX structure. Based on the resultant simulations of the ion trajectories (see Fig. 6C for examples), the conductance is 20 pS at 0.15 M. Please note this value is corrected to 0.15 M salt whereas the previous value cited was at 0.5 M. We have run tests to check that the conductance depends linearly on [NaCl] up to 0.5 M. This value is indeed smaller than the experimental estimates – this may reflect missing aspects of the structure and/or limitations of the simulations. We have indicated this in the text now.

Interestingly, recent discussions at the BPS20 meeting with the authors of poster 2852-Pos (authors Zhuang, Gharpure, Hibbs, Howard, & Lindahl) suggested that simulated conductance estimates of the nAChR were also lower than the experimental estimates. This work is not yet published so we cannot check the details further, but it suggests that the current state of play is such that computational electrophysiology simulations of pLGICs are able to indicate relative but not absolute conductance values. Interestingly, our estimate of the conductance of the open GlyR (i.e. GlyR+PTX) and of the ‘super-open’ (3JAE) states are both lower than conductances reported from other simulations of 3JAE by Gonzalez Gutierrez et al. and by Cerdan et al.. However, we note that these latter studies were of the simulations of the transmembrane domain (TMD) whereas we simulated the “complete” receptor. From simulation studies of TMD vs. intact receptor for both the GlyR and the 5HT3R (data not shown), we estimate that the TMD alone exhibits a 5 to 15x greater simulated conductance than does the intact protein.

Minor:

Page 3, third line from the bottom: “performed” should be removed.

Done.

Page 5, first sentence. The sentence should be rewritten: authors would like to believe that the conditions of their experiment favor physiological conditions but they cannot be so sure of this.

We have removed physiological and reworded the sentence to “We therefore determined high-resolution structures of the full-length GlyR in a membrane nanodisc environment by single-particle cryo-electron microscope (Cryo-EM) under conditions that stabilize various functional states.”

Page 10, last line. There is no panel D in Figure 3.

Typo, fixed.

Why are the initial particle numbers approximate in Suppl. Table 1?

Add the actual particle numbers.

Are the map sharpening B factors correct? They seem very small (in absolute value).

Yes, they are correct.

Reviewer #3 (Remarks to the Author):

In this work, structural basis for gating, ion selectivity, and single-channel conductance of GlyR in membrane was investigated based on the cryo-EM structures and molecular dynamics simulations. I particularly evaluate molecular dynamics simulation parts of this work.

The molecular dynamics simulations were performed with standard computational technique. The comparison between the simulations and the cryo-EM structures is scientifically meaningful. However, there are a number of uncertain points for this reviewer, which should be clarified in a revised manuscript.

Main Comments:

1) In Page 12, the authors suggested that the interactions between subunits were changed, and the overall buried surface area was decreased in the GlyR-Gly/PTX and GlyR-Gly. However, relationship between interaction change and surface area change is not so clearly described. Why did the authors quantify the overall surface area change? And also, what is the "buried surface" area?

Buried surface area refers to the solvent-accessible surface area of monomeric units that gets buried upon assembly formation, in this case a pentamer. Glycine-induced conformational change involves a global twisting motion that is accompanied by an iris-like opening of the transmembrane helices. This type of movement leads to changes in the solvent exposure of regions, particularly at the interfaces between domains and subunits. We used PDBePISA server to calculate buried surface area (or water inaccessible regions – to a 1.4 Å diameter water probe) as a way to assess changes in interaction interfaces. We now state this in the methods section.

2) In Page 17, the authors described that the pore features of the three conformations were stably maintained in the MD simulations without positional restraints. Is the pore still dewetted at Leu9' in Gly-Apo or wetted in GlyR-Gly/PTX and GlyR-Gly, as shown in Figure 6C?

The pore indeed remained de-wetted at Leu 9' in GlyR-Apo and wetted in GlyR-Gly and GlyR-Gly/PTX, corresponding to a clear absence and presence of an energetic barrier in their respective water free energy profiles (previous Supplemental Figure 13). The same findings are expected for our updated models, which have shown little change in their pore profiles. We have removed the sections for unrestrained simulations from the current version.

Minor comments:

1) Typo: In Pages 10 and 11, Fig3D and Fig 3E should be Fig 2D and Fig2E. There is no explanation about Fig2F in the main text.

Done. The Fig 2F panel is now mentioned with the discussion in page 11.

2) Panels in all figures are very busy. For example, there is a clash between Panels A and B in Figure 1. Position of all panels should be rearranged well.

Done.

3) It is difficult to recognize some labels in the figures (e.g., Figure 2C, Panel A (right) in Figure 3, and Figure 4)

We apologize for the image quality. Some of the depth cueing effect did not come out as expected during the PDF conversion. We have fixed this issue now.

3.0 Å  7.0 Å

Reviewer Figure 1. GlyR-Gly/PTX structure from 5 mM Glycine/5 mM PTX condition. A) Side view of the final 3D reconstruction B) Local resolution calculated using RESMAP C) Fourier shell correlation (FSC) curves before (red) and after post-processing (blue) using gold-standard refinement in RELION 3.0 D) Cross validation of model refinement, FSC curves of the refined model versus summed map (full dataset), refined model versus half map 1 (used during refinement), and refined model versus half map 2 (not used during refinement).

Reviewer Figure 2. Alignment of 3D maps of 3 mM (gray) and 5 mM PTX (orange) reconstructions. B) Alignment of structures refined from the two reconstructions. C) Density for M2 and PTX contoured at the same level for the 5 mM PTX data.

REVIEWERS' COMMENTS second round:

Reviewer #1 (Remarks to the Author):

In their revised manuscript Kumar et al. have addressed most of my concerns from the reviews of the two previous versions of the manuscript that I have reviewed.

As for my bigger concerns and changes implemented in response to them:

- The authors have now included mentioning and justification of the approach of using the Gly/PTX-GlyR structure as a surrogate for the open GlyR concentration, so that the reader now is made aware of this issue and can explore and use this particular structure while keeping this in mind.
- The inclusion of the ivermectin-structures in the first version of the manuscript submitted to Nat. Comm. represents a substantial contribution to the study, as they add more information about the conformational transitions of the GlyR and potentially also other pLGICs.
- Although some of the conclusions drawn by the authors based on this study mainly confirm and substantiate findings already extractable from previous GlyR and pLGIC structures, the study does introduce novel information and add to the knowledge about the closed, open and desensitized conformations and the transitions between them.

- When it comes to addressing my concern about the lack of functional verification of the putative interactions between specific residues depicted in the former Figs. 4A and 4B that potentially could be of functional importance, the authors have been less responsive:

As also mentioned in their rebuttal to my comments to a former version of this manuscript, the authors point out that they include references to previous studies showing the functional importance of residues forming network of interactions in the ECD/TMD interface. However, this network is shown in Fig. 3, and not in Fig. 4A and 4B which was the figures my comment was aimed at. In their rebuttal to this version of the manuscript the authors acknowledge (at least I interpret their comment to mean that that) that further functional characterization is needed in order to assess the functional importance of the interactions previously depicted in Fig. 4A and 4B, and they have also revised the two figures so that these putative interactions are now longer shown in them. I acknowledge that this means that the authors no longer claim or hint to these interactions being of functional importance without having investigated (and substantiated) this. While this certainly is a good thing and this version of the manuscript thus is improved on this account compared to the former version, I can not help feeling that the authors are missing a golden opportunity here: to dive further into and use this information provided by their structures and through functional studies translate a potentially interesting observation into solid conclusions about the specific molecular basis underlying the conformational transitions of the GlyR. In their rebuttal to a previous version of this manuscript, the authors argued that such an analysis would be very elaborate, but I would argue that a fairly focused functional characterization of a small series of selected GlyR mutants could provide solid experimental data to either substantiate or disprove a significant functional importance of these interactions.

In conclusion, when it comes to this issue, the authors are no longer vaguely hinting to a possible functional importance of these interactions in the manuscript, but they are not "breathing life" into their structures by investigating the putative importance of these either.

As I also stated as my overall conclusion based on review of the former version of this manuscript, this is a very nice study that provides high-quality structures of the GlyR in its three major conformations as well as additional proposed intermediate conformations using advanced cryo-EM and molecular dynamics techniques. Thus, my overall assessment of the work is that this is a study that will be of considerable interest to the GlyR and pLGIC fields and to the neuroscience and structural biology fields in general.

Anders A. Jensen

Reviewer #2 (Remarks to the Author):

I think authors did good job in addressing all of the critiques and the manuscript is now acceptable for publication in Nature Communications.

Reviewer #3 (Remarks to the Author):

This is my second review on the work. In the previous review, I pointed out two major concerns and three minor ones. In their revised manuscript, they answered my questions and suggestions clearly so that I don't have any more. I basically agree the publication of this work.

Only one concern is that in their revised manuscript, they mentioned that "We have removed the sections for unrestrained simulations from the current version". I don't understand the reason why they have removed them although in the previous manuscript they mentioned that the pore features of the three conformations were stably maintained in the MD simulations without positional restraints. I guess that the resolution of this model might not be enough for unrestrained MD simulations. If so, I would like to suggest the authors to add one sentence for the reliability of unrestrained MD simulation based on the experimental data, if they could in future revision or proof.

REVIEWERS' COMMENTS:

We thank all the three Reviewers for their invaluable time in providing constructive feedback on our manuscript. We believe that these comments and suggestions have significantly improved our manuscript.

Reviewer #1 (Remarks to the Author):

In their revised manuscript Kumar et al. have addressed most of my concerns from the reviews of the two previous versions of the manuscript that I have reviewed.

As for my bigger concerns and changes implemented in response to them:

- The authors have now included mentioning and justification of the approach of using the Gly/PTX-GlyR structure as a surrogate for the open GlyR concentration, so that the reader now is made aware of this issue and can explore and use this particular structure while keeping this in mind.
- The inclusion of the ivermectin-structures in the first version of the manuscript submitted to Nat. Comm. represents a substantial contribution to the study, as they add more information about the conformational transitions of the GlyR and potentially also other pLGICs.
- Although some of the conclusions drawn by the authors based on this study mainly confirm and substantiate findings already extractable from previous GlyR and pLGIC structures, the study does introduce novel information and add to the knowledge about the closed, open and desensitized conformations and the transitions between them.
- When it comes to addressing my concern about the lack of functional verification of the putative interactions between specific residues depicted in the former Figs. 4A and 4B that potentially could be of functional importance, the authors have been less responsive: As also mentioned in their rebuttal to my comments to a former version of this manuscript, the authors point out that they include references to previous studies showing the functional importance of residues forming network of interactions in the ECD/TMD interface. However, this network is shown in Fig. 3, and not in Fig. 4A and 4B which was the figures my comment was aimed at. In their rebuttal to this version of the manuscript the authors acknowledge (at least I interpret their comment to mean that that) that further functional characterization is needed in order to assess the functional importance of the interactions previously depicted in Fig. 4A and 4B, and they have also revised the two figures so that these putative interactions are now longer shown in them. I acknowledge that this means that the authors no longer claim or hint to these interactions being of functional importance without having investigated (and substantiated) this. While this certainly is a good thing and this version of the manuscript thus is improved on this account compared to the former version, I can not help feeling that the authors are missing a golden opportunity here: to dive further into and use this information provided by their structures and through functional studies translate a potentially interesting observation into solid conclusions about the specific molecular basis underlying the conformational transitions of the GlyR. In their rebuttal to a previous version of this manuscript,

the authors argued that such an analysis would be very elaborate, but I would argue that a fairly focused functional characterization of a small series of selected GlyR mutants could provide solid experimental data to either substantiate or disprove a significant functional importance of these interactions. In conclusion, when it comes to this issue, the authors are no longer vaguely hinting to a possible functional importance of these interactions in the manuscript, but they are not “breathing life” into their structures by investigating the putative importance of these either.

As I also stated as my overall conclusion based on review of the former version of this manuscript, this is a very nice study that provides high-quality structures of the GlyR in its three major conformations as well as additional proposed intermediate conformations using advanced cryo-EM and molecular dynamics techniques. Thus, my overall assessment of the work is that this is a study that will be of considerable interest to the GlyR and pLGIC fields and to the neuroscience and structural biology fields in general.

Thanks for the positive comments. We do agree with the Reviewer on this point and an in depth characterization of GlyR structure-function is currently underway.

Reviewer #2 (Remarks to the Author):

I think authors did good job in addressing all of the critiques and the manuscript is now acceptable for publication in Nature Communications.

Thank you, we are pleased to note that the Reviewer finds the revision satisfactory.

Reviewer #3 (Remarks to the Author):

This is my second review on the work. In the previous review, I pointed out two major concerns and three minor ones. In their revised manuscript, they answered my questions and suggestions clearly so that I don't have any more. I basically agree the publication of this work.

Only one concern is that in their revised manuscript, they mentioned that "We have removed the sections for unrestrained simulations from the current version". I don't understand the reason why they have removed them although in the previous manuscript they mentioned that the pore features of the three conformations were stably maintained in the MD simulations without positional restraints. I guess that the resolution of this model might not be enough for unrestrained MD simulations. If so, I would like to suggest the authors to add one sentence for the reliability of unrestrained MD simulation based on the experimental data, if they could in future revision or proof.

Thank you! In response to the request from this Reviewer we have included the following line that captures the finding from our unrestrained simulation. “Unrestrained simulations (3 x 200 ns) indicate that the GlyR-Gly/PTX is stable, remaining open and presenting no free energy barrier to water.”